# TWIST1 and chromatin regulatory proteins interact to guide neural crest cell differentiation

Xiaochen Fan[1,2†]*, V Pragathi Masamsetti[1], Jane QJ Sun[1], Kasper Engholm-Keller[3‡], Pierre Osteil[1], Joshua Studdert[1], Mark E Graham[3], Nicolas Fossat[1,2§]*, Patrick PL Tam[1,2]*

[1]Embryology Unit, Children's Medical Research Institute, The University of Sydney, Sydney, Australia; [2]The University of Sydney, School of Medical Sciences, Faculty of Medicine and Health, Sydney, Australia; [3]Synapse Proteomics Group, Children's Medical Research Institute, The University of Sydney, Sydney, Australia

**\*For correspondence:**
x6fan@eng.ucsd.edu (XF);
nfossat@sund.ku.dk (NF);
PTam@cmri.org.au (PPLT)

**Present address:** [†]Department of Bioengineering, University of California, San Diego, United States; [‡]Department of Food Science, University of Copenhagen, Copenhagen, Denmark; [§]Copenhagen Hepatitis C Program (CO-HEP), Department of Immunology and Microbiology, University of Copenhagen, and Department of Infectious Diseases, Hvidovre Hospital, Copenhagen, Denmark

**Competing interests:** The authors declare that no competing interests exist.

**Abstract** Protein interaction is critical molecular regulatory activity underlining cellular functions and precise cell fate choices. Using TWIST1 BioID-proximity-labeling and network propagation analyses, we discovered and characterized a TWIST-chromatin regulatory module (TWIST1-CRM) in the neural crest cells (NCC). Combinatorial perturbation of core members of TWIST1-CRM: TWIST1, CHD7, CHD8, and WHSC1 in cell models and mouse embryos revealed that loss of the function of the regulatory module resulted in abnormal differentiation of NCCs and compromised craniofacial tissue patterning. Following NCC delamination, low level of TWIST1-CRM activity is instrumental to stabilize the early NCC signatures and migratory potential by repressing the neural stem cell programs. High level of TWIST1 module activity at later phases commits the cells to the ectomesenchyme. Our study further revealed the functional interdependency of TWIST1 and potential neurocristopathy factors in NCC development.

## Introduction

The cranial neural crest cell (NCC) lineage originates from the neuroepithelium (*Vokes et al., 2007*; *Groves and LaBonne, 2014*; *Mandalos and Remboutsika, 2017*) and contributes to the craniofacial tissues in vertebrates (*Sauka-Spengler and Bronner-Fraser, 2008*) including parts of the craniofacial skeleton, connective tissues, melanocytes, neurons, and glia (*Kang and Svoboda, 2005*; *Blentic et al., 2008*; *Ishii et al., 2012*; *Theveneau and Mayor, 2012*). The development of these tissues is affected in neurocristopathies, which can be traced to mutations in genetic determinants of NCC specification and differentiation (*Etchevers et al., 2019*). As an example, mutations in transcription factor *TWIST1* in human are associated with craniosynostosis (*El Ghouzzi et al., 2000*) and cerebral vasculature defects (*Tischfield et al., 2017*). Phenotypic analyses of *Twist1* conditional knockout mouse revealed that TWIST1 is required in the NCCs for the formation of the facial skeleton, the anterior skull vault, and the patterning of the cranial nerves (*Soo et al., 2002*; *Ota et al., 2004*; *Bildsoe et al., 2009*; *Bildsoe et al., 2016*). To comprehend the mechanistic complexity of NCC development and its implication in a range of diseases, it is essential to collate the compendium of genetic determinants of the NCC lineage and characterize how they act in concert in time and space.

During neuroectoderm development, transcriptional programs are initiated successively in response to morphogen induction to specify neural stem cell (NSC) subdomains along the dorsal-ventral axis in the neuroepithelium (*Briscoe et al., 2000*; *Vokes et al., 2007*; *Kutejova et al., 2016*). NCCs also arise from the neuroepithelium, at the border with the surface ectoderm through the pre-

**eLife digest** Shaping the head and face during development relies on a complex ballet of molecular signals that orchestrates the movement and specialization of various groups of cells. In animals with a backbone for example, neural crest cells (NCCs for short) can march long distances from the developing spine to become some of the tissues that form the skull and cartilage but also the pigment cells and nervous system.

NCCs mature into specific cell types thanks to a complex array of factors which trigger a precise sequence of binary fate decisions at the right time and place. Amongst these factors, the protein TWIST1 can set up a cascade of genetic events that control how NCCs will ultimately form tissues in the head. To do so, the TWIST1 protein interacts with many other molecular actors, many of which are still unknown.

To find some of these partners, Fan et al. studied TWIST1 in the NCCs of mice and cells grown in the lab. The experiments showed that TWIST1 interacted with CHD7, CHD8 and WHSC1, three proteins that help to switch genes on and off, and which contribute to NCCs moving across the head during development. Further work by Fan et al. then revealed that together, these molecular actors are critical for NCCs to form cells that will form facial bones and cartilage, as opposed to becoming neurons. This result helps to show that there is a trade-off between NCCs forming the face or being part of the nervous system.

One in three babies born with a birth defect shows anomalies of the head and face: understanding the exact mechanisms by which NCCs contribute to these structures may help to better predict risks for parents, or to develop new approaches for treatment.

epithelial-mesenchymal transition (pre-EMT) which is marked by the activation of _Tfap2a, Id1, Id2, Zic1, Msx1_ and _Msx2_ (_Baker et al., 1997_; _Mayor et al., 1997_; _Saint-Jeannet et al., 1997_; _Marchant et al., 1998_; _Etchevers et al., 2019_). In the migratory NCCs, gene activity associated with pre-EMT and NCC specification is replaced by that of EMT and NCC identity (_Marchant et al., 1998_). NCC differentiation progresses in a series of cell fate decisions (_Lasrado et al., 2017_; _Soldatov et al., 2019_). Genetic activities for mutually exclusive cell fates are co-activated in the progenitor population, which is followed by an enhancement of the transcriptional activities that predilect one lineage over the others (_Lasrado et al., 2017_; _Soldatov et al., 2019_). However, more in-depth knowledge of the specific factors triggering this sequence of events and cell fate bias is presently lacking.

_Twist1_ expression is initiated during NCC delamination and its activity is sustained in migratory NCCs to promote ectomesenchymal fate (_Soldatov et al., 2019_). TWIST1 mediates cell fate choices through functional interactions with other basic-helix-loop-helix (bHLH) factors (_Spicer et al., 1996_; _Firulli et al., 2005_; _Connerney et al., 2006_) in addition to transcription factors SOX9, SOX10, and RUNX2 (_Spicer et al., 1996_; _Hamamori et al., 1997_; _Bialek et al., 2004_; _Laursen et al., 2007_; _Gu et al., 2012_; _Vincentz et al., 2013_). TWIST1 therefore constitutes a unique assembly point to identify the molecular modules necessary for cranial NCC development and determine how they orchestrate the sequence of events in this process.

To decipher the molecular context of TWIST1 activity and identify functional modules, we generated the first TWIST1 protein interactome in the NCCs. Leveraging the proximity-dependent biotin identification (BioID) methodology, we captured TWIST1 interactions in the native cellular environment including previously intractable transient and low-frequency events which feature interactions between transcription regulators (_Roux et al., 2012_; _Kim and Roux, 2016_). Integrating prior knowledge of protein associations and applying network propagation analysis (_Cowen et al., 2017_), we uncovered modules of highly connected interactors as potent NCC regulators. Among the top-ranked candidates were histone modifiers and chromatin remodelers that constitute the functional chromatin regulatory module (TWIST1-CRM) in NCC. Genome occupancy, gene expression, and combinatorial perturbation studies of high-ranked members of the TWIST1-CRM during neurogenic differentiation in vitro and in embryos revealed their necessity in stabilizing the identity of early migratory NCC and subsequent acquisition of ectomesenchyme potential. This study also

highlighted the concurrent activation and cross-repression of the molecular machinery that governs the choice of cell fates between neural crest and neurogenic cell lineages.

## Results

### Deciphering the TWIST1 protein interactome in cranial NCCs using BioID

The protein interactome of TWIST1 was characterized using the BioID technique which allows for the identification of interactors in their native cellular environment (*Figure 1A*). We performed the experiment in cranial NCC cell line O9-1 (*Ishii et al., 2012*) transfected with TWIST1-BirA* (TWIST1 fused to the BirA* biotin ligase). In the transfected cells, biotinylated proteins were predominantly

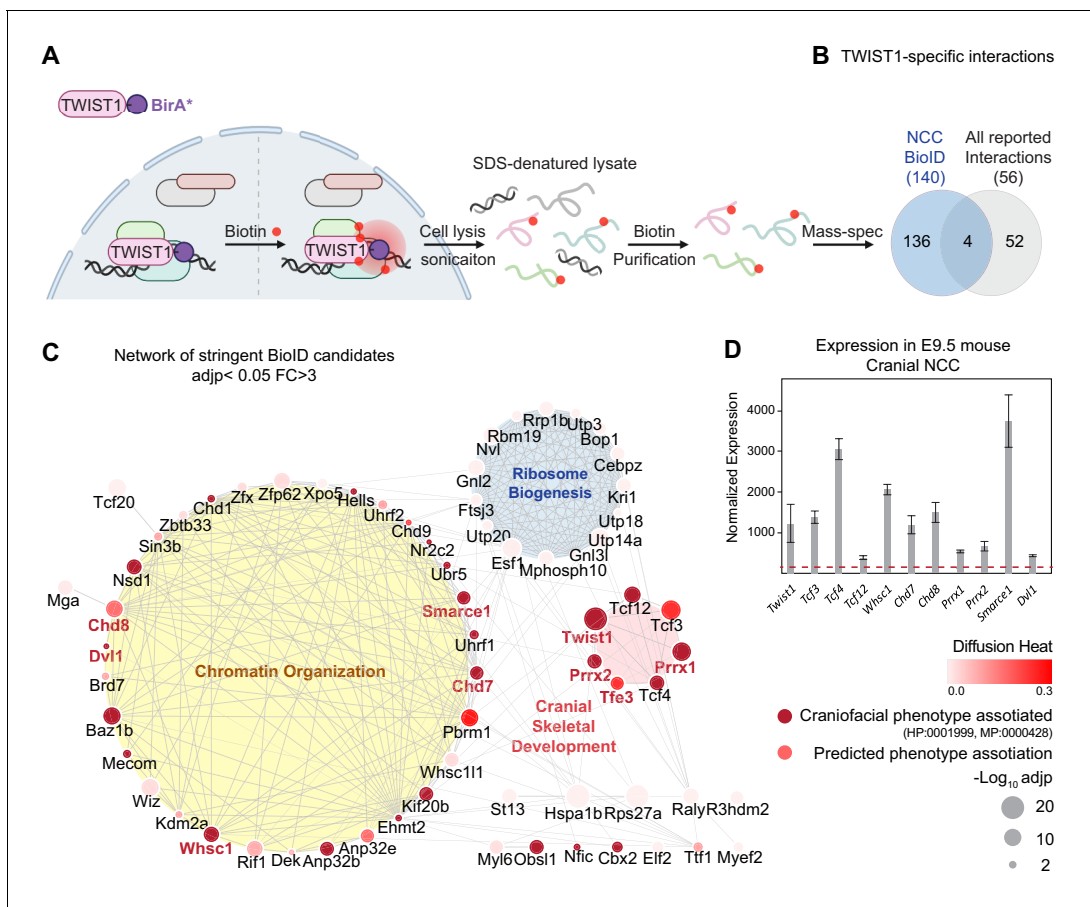

**Figure 1.** TWIST1 interactome in cranial NCCs revealed using BioID and network propagation. (A) BioID procedure to identify TWIST1-interacting partners in neural crest stem cells (NCCs). TWIST1-BirA* (TWIST1 fused to the BirA* biotin ligase) labeled the proteins partners within the 10 nm proximity in live cells. Following cell lysis and sonication, streptavidin beads were used to capture denatured biotin-labeled proteins, which were purified and processed for mass spectrometry analysis. (B) TWIST1-specific interaction candidates identified by BioID mass-spectrometry analysis in NCC cell line (p<0.05; Fold-change >3; PSM#>2) overlap with all reported TWIST1 interactions on the Agile Protein Interactomes DataServer (APID) (*Alonso-López et al., 2019*). (C) Networks constructed from stringent TWIST1-specific interaction at a significant threshold of adjusted p-value (adjp) <0.05 and Fold-change >3. Unconnected nodes were removed. Top GO terms for proteins from three different clusters are shown. Node size = -Log10 (adjp). Genes associated with human and mouse facial malformation (HP:0001999, MP:0000428) were used as seeds (dark red) for heat diffusion through network neighbors. Node color represents the heat diffusion score. (D) Expression of candidate interactor genes in cranial neural crest from E9.5 mouse embryos; data were derived from published transcriptome dataset (*Fan et al., 2016*). Each bar represents mean expression ± SE of three biological replicates. All genes shown are expressed at level above the microarray detection threshold ($2^7$, red dashed line).

The online version of this article includes the following figure supplement(s) for figure 1:

**Figure supplement 1.** Nuclear localization of TWIST1-BirA* biotinylated proteins and the endogenous TWIST1.

**Figure supplement 2.** Identification of core NCC regulators within the TWIST1-CRM.

localized in the nucleus (*Figure 1—figure supplement 1A,B*; *Singh and Gramolini, 2009*). The profile of TWIST1-BirA* biotinylated proteins were different from that of biotinylated proteins captured by GFP-BirA* (*Figure 1—figure supplement 2A*). Western blot analysis detected TCF4, a known dimerization partner of TWIST1, among the TWIST1-BirA* biotinylated proteins but not in the control group (*Figure 1—figure supplement 2A*). These findings demonstrated the utility and specificity of the BioID technology to identify TWIST1-interacting proteins.

We characterized all the proteins biotinylated by TWIST1-BirA* and GFP-BirA* followed by streptavidin purification using liquid chromatography combined with tandem mass spectrometry (LC-MS/MS) (*Supplementary file 1*). Differential binding analysis of TWIST1 using sum-normalized peptide-spectrum match (PSM) values (*Figure 1—figure supplement 2B,C*; see Materials and methods) revealed 140 putative TWIST1 interactors in NCCs (p<0.05; Fold-change >3; PSM#>2; *Figure 1B*, *Supplementary file 1*). These candidates included 4 of 56 known TWIST1 interactors, including TCF3, TCF4, TCF12, and GLI3 (overlap odds ratio = 18.05, Chi-squared test p-value=0.0005; Agile Protein Interactomes DataServer [APID]) (*Alonso-López et al., 2019*; *Fan et al., 2020*). Despite that the APID database covers a broad spectrum of protein interaction in different cell line models, it was noted that the TWIST1 partners, TCF3, 4, and 12 that were recurrently identified in yeast-two-hybrid, immunoprecipitation and in vitro interaction assays were recovered by BioID (*El Ghouzzi et al., 2000*; *Firulli et al., 2007*; *Fu et al., 2011*; *Teachenor et al., 2012*; *Sharma et al., 2013*; *Kotlyar et al., 2015*; *Li et al., 2015*). This finding has prompted us to explore the rest of the novel partners identified in the BioID analysis.

## Network propagation prioritized functional modules and core candidates in TWIST1 interactome

We invoked network propagation analytics to identify functional modules amongst novel TWIST1 BioID-interactors and to prioritize the key NCC regulators (see Materials and methods). Network propagation, which is built on the concept of 'guilt-by-association', is a set of analytics used for gene function prediction and module discovery (*Sharan et al., 2007*; *Ideker and Sharan, 2008*; *Cowen et al., 2017*). By propagating molecular and phenotypic information through connected neighbors, this approach identified and prioritized relevant functional clusters while eliminating irrelevant ones.

The TWIST1 functional interaction network was constructed by integrating the association probability matrix of the BioID candidates based on co-expression, protein-interaction, and text mining databases from STING (*Singh and Gramolini, 2009*; *Szklarczyk et al., 2015*). Markov clustering (MCL) was applied to the matrix for the inference of functional clusters (*Figure 1—figure supplement 2D*, *Supplementary file 2*). Additionally, data from a survey of the interaction of 56 transcription factors and 70 unrelated control proteins were used to distinguish the likely specific interactors from the non-specific and the promiscuous TF interactors (*Li et al., 2015*). Specific TF interactors (red) and potential new interactors (blue; *Figure 1—figure supplement 2D–i*) clustered separately from the hubs predominated by non-specific interactors (gray; *Figure 1—figure supplement 2D–ii*). The stringency of the screen was enhanced by increasing the statistical threshold (adjusted p-value [adjp]<0.05) and excluding the clusters formed by non-specific interactors such as those containing heat-shock proteins and cytoskeleton components. Gene Ontology analysis revealed major biological activities of proteins in the clusters: chromatin organization, cranial skeletal development, and ribosome biogenesis (*Figure 1C*; *Supplementary file 2*; *Chen et al., 2009*).

Heat diffusion was applied to prioritize key regulators of NCC development. The stringent TWIST1 interaction network comprises proteins associated with facial malformation phenotypes in human/mouse (HP:0001999, MP:0000428), which points to a likely role in NCC development. These factors were used as seeds for a heat diffusion simulation to find near-neighbors of the phenotype hot-spots (i.e. additional factors that may share the phenotype) and to determine their hierarchical ranking (*Figure 1C*, *Supplementary file 2*). As expected, that disease causal factors are highly connected and tend to interact with each other (*Jonsson and Bates, 2006*), a peak of proteins with high degrees of connectivity emerged among the top diffusion ranked causal factors, most of which are from the chromatin organization module (*Figure 1—figure supplement 2F*). TWIST1 and these interacting chromatin regulators were referred to hereafter as the TWIST1-chromatin regulatory module (TWIST1-CRM).

Among the top 30 diffusion ranked BioID candidates, we prioritized nine for further characterization. These included chromatin regulators that interact with TWIST1 exclusively in NCCs versus 3T3 fibroblasts: the chromodomain helicases CHD7, CHD8, the histone methyltransferase WHSC1 and SMARCE1, a member of the SWI/SNF chromatin remodeling complex (*Figure 1C*, candidates name in red; *Figure 1—figure supplement 2E,F*; *Supplementary file 3*). We also covered other types of proteins, including transcription factors PRRX1, PRRX2, TFE3 and the cytoplasmic phosphoprotein DVL1 (Dishevelled 1). The genes encoding these proteins were found to be co-expressed with *Twist1* in the cranial NCCs of in embryonic day (E) 9.5 mouse embryos (*Figure 1D*, *Supplementary file 1*; *Bildsoe et al., 2016*; *Fan et al., 2016*).

## The chromatin regulators interact with the N-terminus domain of TWIST1

Co-immunoprecipitation (co-IP) assays showed that CHD7, CHD8, PRRX1, PRRX2, and DVL1 could interact with TWIST1 like known interactors TCF3 and TCF4, while TFE3 and SMARCE1 did not show any detectable interaction (*Figure 2A*). Fluorescent immunostaining demonstrated that these proteins co-localized with TWIST1 in the nucleus (*Figure 1—figure supplement 1C*). The exceptions were DVL1 and TFE3, which were localized predominantly in the cytoplasm (*Figure 1—figure supplement 1C*). Among these candidates, CHD7 and CHD8 are known to engage in direct domain-specific protein-protein interactions (*Batsukh et al., 2010*). Three sub-regions of CHD7 and CHD8 were tested for interaction with TWIST1 (*Figure 2B*). For both proteins, the p1 region, which encompasses helicases and chromodomains, showed no detectable interaction with partial or full-length TWIST1. In contrast, the p2 and the p3 regions of CHD7 and CHD8 interacted with full-length TWIST1 as well as with its N-terminal region (*Figure 2C*). Reciprocally, the interaction was tested with different regions of TWIST1 including the bHLH domain, the WR domain, the C-terminal region and the N-terminal region (*Figure 2B*). CHD7, CHD8, and WHSC1 interacted preferentially with the TWIST1 N-terminus whereas the TCF dimerization partners interacted specifically with the bHLH domain (*Figure 2D*). Consistent with the co-IP result, SMARCE1 and TFE3 did not interact with TWIST1. Interestingly, the other known factor that binds the TWIST1 N-terminal region is the histone acetyltransferase CBP/P300 which is also involved in chromatin remodeling (*Hamamori et al., 1999*). These findings demonstrated the direct interaction of TWIST1 with a range of epigenetic factors and transcriptional regulators and identified the TWIST1 N-terminal region as the domain of contact.

## Genetic interaction of *Twist1* and chromatin regulators in craniofacial morphogenesis

The function of the core components of the TWIST1-CRM was investigated in vivo using mouse embryos derived from ESCs that carried single-gene or compound heterozygous mutations of *Twist1* and the chromatin regulators. Mutant ESCs for *Twist1* and the three validated NCC-exclusive chromatin regulatory partners *Chd7*, Chd8, *and Whsc1* were generated by CRISPR-Cas9 editing (*Figure 3—figure supplement 1A,B*; *Ran et al., 2013*). ESCs of specific genotype (non-fluorescent) were injected into 8 cell host wild-type embryos (expressing fluorescent DsRed.t3) and chimeras were collected at E9.5 or E11.5 (*Figure 3A*; *Sibbritt et al., 2019*).

Only embryos with predominant contribution of mutant ESCs, indicated by the absence or low level of DsRed.t3 fluorescence, were analyzed. The majority of embryos derived from single-gene heterozygous ESCs (*Twist1$^{+/-}$*, *Chd8$^{+/-}$*, and *Whsc1$^{+/-}$*) displayed mild deficiency in the cranial neuroepithelium and focal vascular hemorrhage (*Figure 3B,C*). Compound heterozygous embryos (*Twist1$^{+/-}$;Chd7$^{+/-}$*, *Twist1$^{+/-}$;Chd8$^{+/-}$*, and *Twist1$^{+/-}$;Whsc1$^{+/-}$*) displayed more severe craniofacial abnormalities and exencephaly (*Figure 3B,C*).

Given that CHD8 was not previously known to involve in craniofacial development of the mouse embryo, we focused on elucidating the impact of genetic interaction of *Chd8* and *Twist1* on NCC development in vivo. While *Chd8$^{+/-}$* embryos showed incomplete neural tube closure, compound *Twist1$^{+/-}$;Chd8$^{+/-}$* embryos displayed expanded neuroepithelium, a phenotype not observed in the single-gene mutants (*Figure 3B,E*). The population of NCCs expressing Tfap2α, a Twist1-independent NCC marker (*Brewer et al., 2004*) was reduced in the frontonasal tissue and the trigeminal ganglion (*Figure 3E–i,F*). In contrast, SOX2 expression was upregulated in the ventricular zone of the neuroepithelium of mutant chimeras (*Figure 3E–ii,iii,G*). Furthermore, *Twist1$^{+/-}$*, *Chd8$^{+/-}$* and

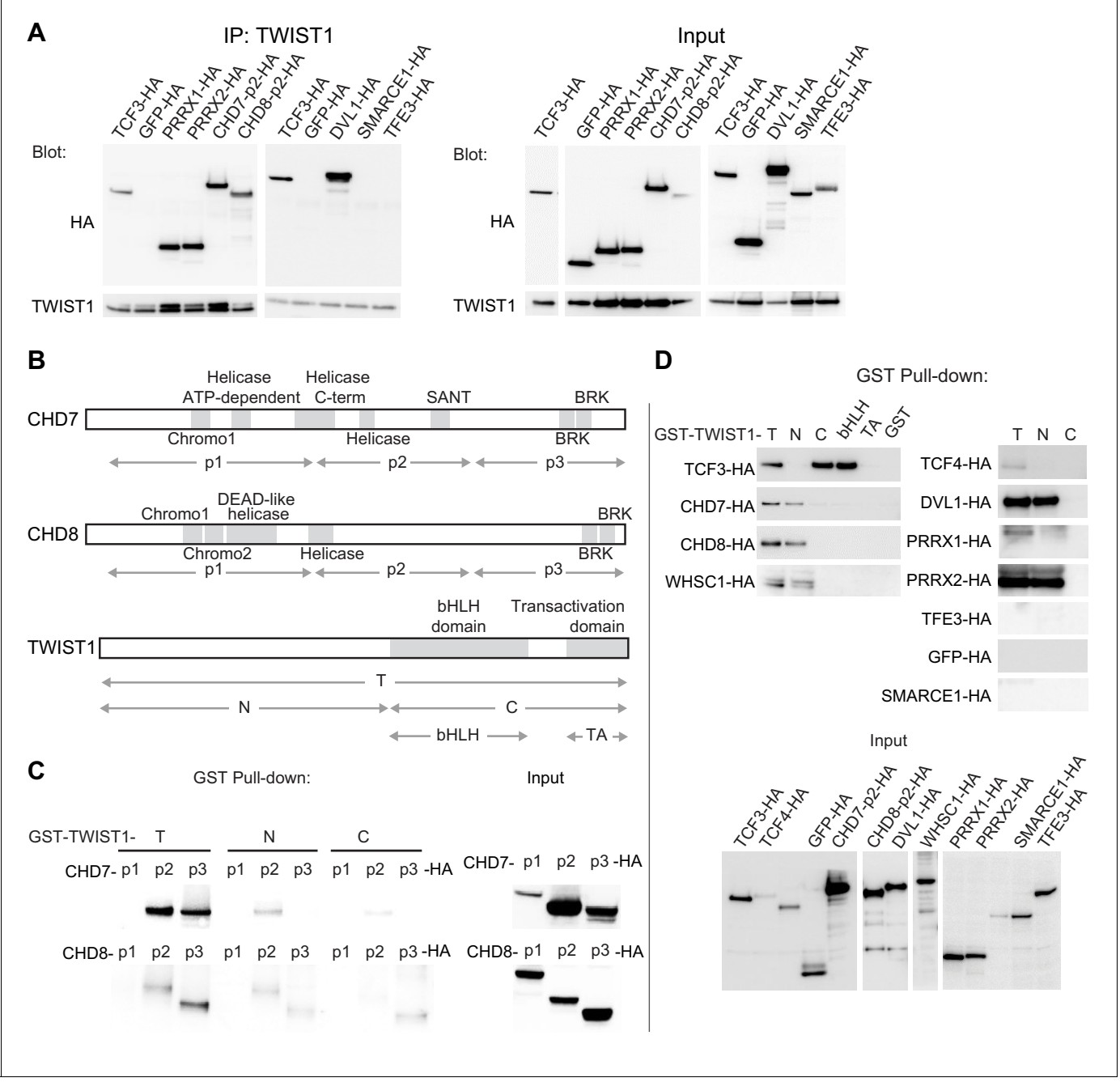

**Figure 2.** The chromatin regulators interact with the N-terminus domain of TWIST1. (**A**) Detection of HA-tagged proteins after immunoprecipitation (IP) of TWIST1 (IP: α-TWIST1) from lysates of O9-1 cells transfected with constructs expressing TWIST1 (input blot: α-TWIST1) and the HA-tagged proteins partners (input blot: α-HA). (**B**) Schematics of CHD7, CHD8, and TWIST1 proteins showing the known domains (gray blocks) and the regions (double arrows) tested in the experiments shown in panels C and D. (**C, D**) Western blot analysis of HA-tagged proteins (α-HA antibody) after GST-pulldown with different TWIST1 domains (illustrated in B). Protein expression in the input is displayed separately. T, full-length TWIST1; N, N-terminal region; C, C-terminal region; bHLH, basic helix-loop-helix domain; TA, transactivation domain.

$Twist1^{+/-};Chd8^{+/-}$ embryos displayed different degrees of hypoplasia of the NCC-derived cranial nerves (*Figure 3H*). Cranial nerves III and IV were absent, and nerve bundle in the trigeminal ganglia showed reduced thickness, most evidently in the $Twist1^{+/-};Chd8^{+/-}$ compound mutant embryos (*Figure 3H,I*).

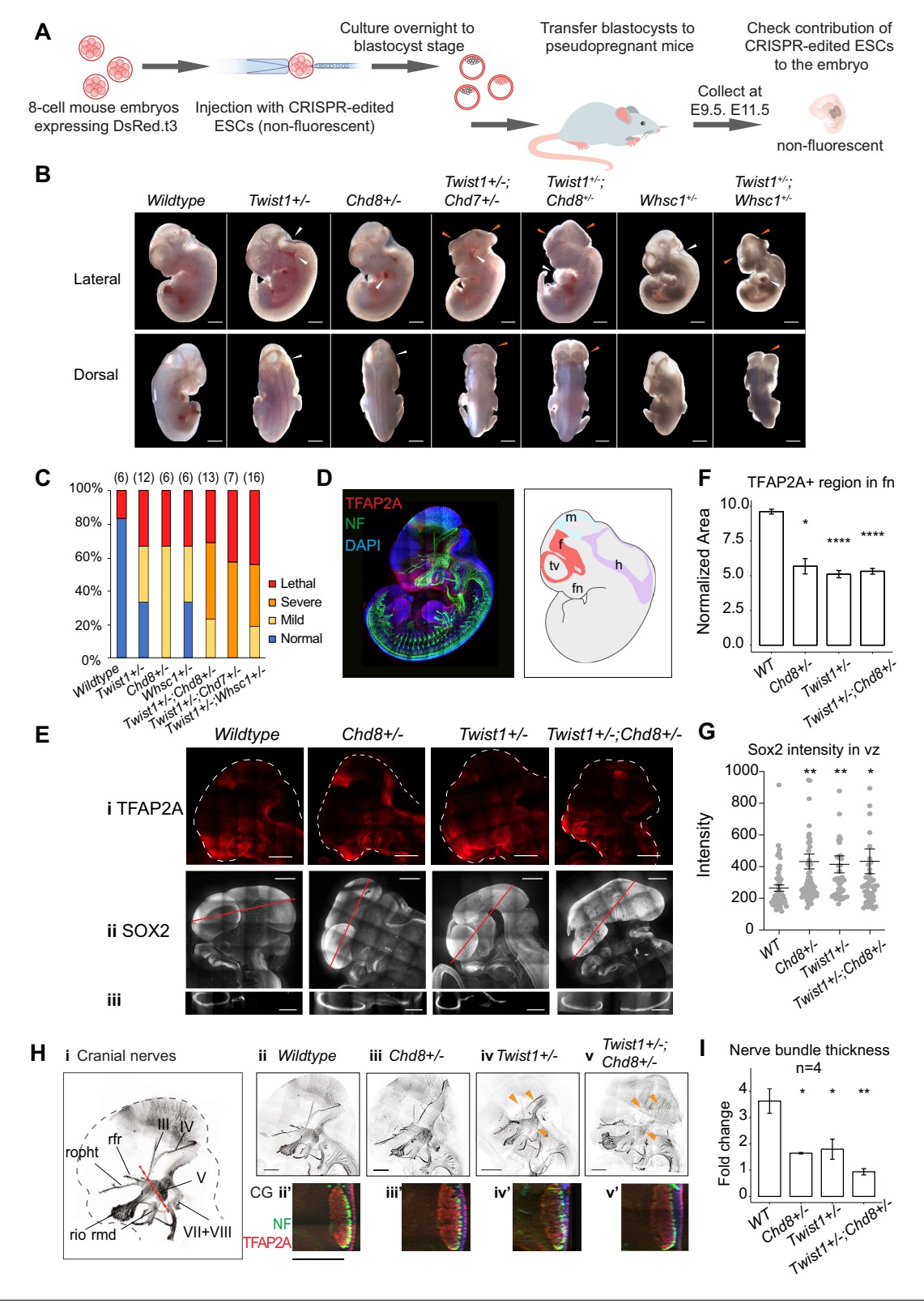

**Figure 3.** Genetic interaction of Twist1 and chromatin regulators in craniofacial morphogenesis. (A) Experimental strategy for generating chimeric mice from WT and mutant ESCs (see Materials and methods). (B) Lateral and dorsal view of mid-gestation chimeric embryos with predominant ESC contribution (embryos showing low or undetectable red fluorescence). Genotype of ESC used for injection is indicated. Scale bar: 1 mm. Heterozygous embryos of single genes (*Twist1*[+/-], *Chd8*[+/-], *Whsc1*[+/-]) showed mild defects including hemorrhages and mild neural tube defect (white arrowheads). *Figure 3 continued on next page*

*Figure 3 continued*

Compound heterozygous embryos displayed open neural tube and head malformation (orange arrowheads, n ≥ 6 for each genotype, see panel 3C), in addition to heart defects. (C) Proportions of normal and malformed embryos (Y-axis) for each genotype (X-axis). Severity of mutant phenotypes was determined based on the incidence of developmental defects in the neuroepithelium, midline tissues, heart and vasculature: Normal (no defect); Mild (1–2 defects); Severe (3–4 defects), and early lethality. The number of embryos scored for each genotype is in parentheses. (D) Whole-mount immunofluorescence of E11.5 chimeras derived from wildtype ESCs, shows the expression of TFAP2A (red) and neurofilament (NF, green) and cell nuclei by DAPI (blue). Schematic on the right shows the neuroepithelium structures: f, forebrain; m, midbrain; h, hindbrain; tv, telencephalic vesicle; fn, frontonasal region. (E) (i) NCC cells, marked by TFAP2A, and neuroepithelial cells, marked by SOX2, are shown in (ii) sagittal, and (iii) transverse view of the craniofacial region (red line in ii: plane of section). (F) Quantification of frontal nasal TFAP2A+ tissues (mean normalized area ± SE) of three different sections of embryos of each genotype. (G) SOX2 intensity (mean ± SE) in the ventricular zone of three sections of embryos of each genotype were quantified using IMARIS. (H) (i) Cranial nerves visualized by immunostaining of neurofilament (NF). (ii–v) Maximum projection of cranial nerves in embryos. Missing or hypoplastic neurites are indicated by arrowheads. (ii′–v′) Cross-section of neurofilament bundles in the trigeminal ganglion. Red dashed line in i: plane of section. Bar: 500 μm; V, trigeminal ganglion; III, IV, VII, VIII; rio, infraorbital nerve of V2; rmd, mandibular nerve; ropht, ophthalmic profundal nerve of V1; rfr, frontal nerve. (I) Thickness of neural bundle in the trigeminal ganglion was measured by the GFP-positive area, normalized against area of the trigeminal ganglion (TFAP2A+). Values plotted represent mean fold change ± SE. Each condition was compared to *WT*. p-Values were computed by one-way ANOVA. *p<0.05, **p<0.01, ***p<0.001, ****p<0.0001. ns, not significant.

The online version of this article includes the following figure supplement(s) for figure 3:

**Figure supplement 1.** Characterization of CRISPR knockout clones and siRNA knockdown efficiency.

Altogether, these results suggested that TWIST1 genetically interaction with epigenetic regulators CHD7, CHD8, and WHSC1 to guide the formation of the cranial NCC and downstream tissue genesis in vivo.

## Genomic regions co-bound by TWIST1 and chromatin regulators are enriched for early migratory NCC signatures in the open chromatin region

The loss of NCC progenitors and neural tube malformation indicate that the combined activity of TWIST1-chromatin regulators might be required from early in NCC development. To understand the molecular function of TWIST1-chromatin regulators in early NCC differentiation, we performed an integrative ChIP-seq analysis. The ChIP-seq dataset for TWIST1 was generated from the ESC-derived neuroepithelial cells (NECs) which are the source of early NCCs (*Figure 4—figure supplement 1* and Materials and methods). We retrieved published NEC ChIP-seq datasets for CHD7 and CHD8 and the histone modifications and reanalyzed the data following the ENCODE pipeline (*ENCODE Project Consortium, 2012*; *Sugathan et al., 2014*; *Ziller et al., 2015*; *Figure 4—figure supplement 1A*). Two H3K36me3 ChIP-seq datasets for NECs were included in the analysis (*Du et al., 2017*; *Chai et al., 2018*) on the basis that WHSC1 trimethyl transferase targets several H3 lysine (*Morishita et al., 2014*) and catalyzes H3K36me3 modification in vivo (*Nimura et al., 2009*).

Genome-wide co-occupancies of TWIST1, CHD7, and CHD8 showed significant overlap (Fisher's exact test) and clustered by Jaccard Similarity matrix (*Figure 4A*). ChIP-seq peaks were correlated with active histone modifications H3K27ac and H3K4me3 but not the inactive H3K27me3, or the WHSC1-associated H3K36me3 modifications (*Figure 4A*). TWIST1, CHD7, and CHD8 shared a significant number of putative target genes (*Figure 4B*). TWIST1 shared 63% of target genes with CHD8 (odds ratio = 16.93, Chi-squared test p-value<2.2e-16) and 18% with CHD7 (odds ratio = 8.26, p-value<2.2e-16; *Figure 4B*; *Supplementary file 4*). Compared with genomic regions occupied by only one factor, greater percentage of regions with peaks for two or all three factors (TWIST1, CHD7, and CHD8) showed H3K27ac and H3K4me3 signal (*Figure 4C*). This trend was not observed for the H3K27me3 modification. Similarly, the co-occupied transcription start sites (TSS) showed open chromatin signatures with enrichment of H3K4me3 and H3K27Ac and depletion of H3K27me3 (*Ernst et al., 2011*; *Rada-Iglesias et al., 2011*; *Figure 4D,E*). We also did not observe H3K36me3 modifications near the overlapping TSSs, suggesting that WHSC1 may have alternative histone lysine specificity in the NECs.

The top Gene Ontology enriched for the co-occupied regulatory regions of two or three core components included neural tube patterning, cell migration and BMP signaling pathway (*Figure 4F*). Overlapping peaks of the partners were localized within ± 1 kb of the TSS of common target genes

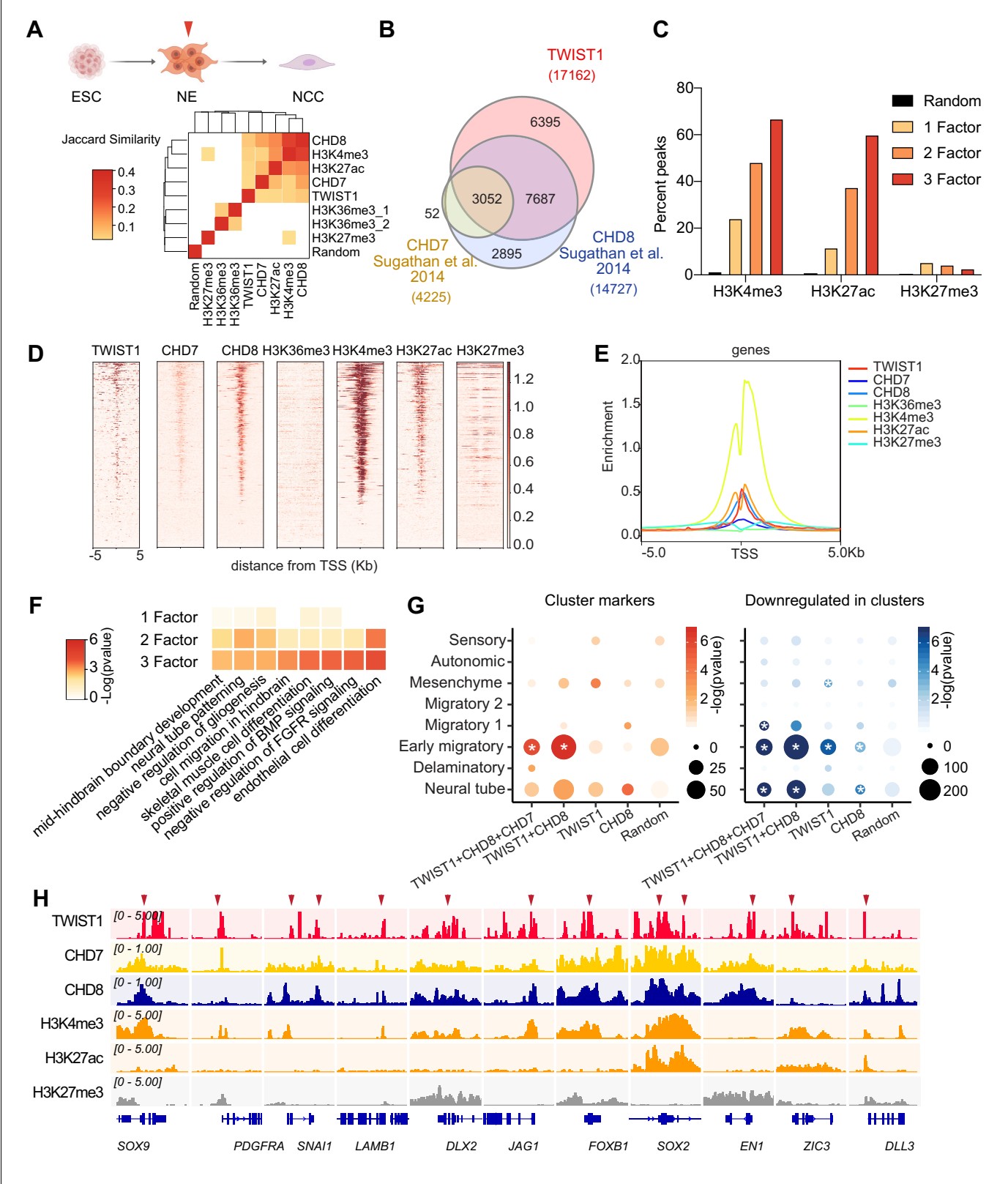

**Figure 4.** Genomic region showing overlapping binding of TWIST1 and partners are enriched for active regulatory signatures and neural tube patterning genes. (A) Top panel: Trajectory of ESC differentiation to neuroepithelial cells (NECs) and NCCs. Bottom panel: Jaccard Similarity matrix generated of ChIP-seq data of TWIST1, CHD7, CHD8, and histone modifications from NE cells. The Jaccard correlation is represented by a color scale.

*Figure 4 continued on next page*

Figure 4 continued

White squares indicate no significant correlation (p<0.05, fisher's exact test) or odds ratio <10 between the two datasets. (B) Venn diagram showing overlaps of putative direct targets of TWIST1, CHD7, and CHD8, based on ChIP-seq datasets for NECs (*Sugathan et al., 2014*). (C) Percent genomic region that is marked by H3K4me3, H3K27ac, and H3K27me3 among regions bound by one, two or all three factors among TWIST1, CHD7, and CHD8. Randomized peak regions of similar length (1 kb) were generated for hg38 as a control. (D) Heatmaps of genomic footprint of protein partners at ± 5 kb from the TSS, based on the ChIP-seq datasets (as in A) and compared with histone marks H3K4me3, H3K27ac, and H3K27me3 in human neural progenitor cells (*Ziller et al., 2015*). TSS lanes with no overlapping signals were omitted. (E) ChIP-seq density profile (rpkm normalized) for all TSS flanking regions shown in D. (F) Gene Ontology analysis of genomic regulatory regions by annotations of the nearby genes. Regions were grouped by presence of binding site of individual factor (TWIST1, CHD7, and CHD8), or by 2–3 factors in combination. The top non-redundant developmental processes or pathways for combinatorial binding peaks or individual factor binding peaks are shown. p-Value cut-off: 0.05. (G) Enrichment of TWIST-chromatin regulator targets among regulons of different NCC single-cell clusters at E8.5-E10.5 (*Soldatov et al., 2019*). Number of overlapping genes with DNA binding peaks (TSS ± 1 kb) for each TF combination are represented by dot size and -log(p-value) is represented by color gradient. Gene modules with significant enrichment (p<0.05) are labeled with asterisk. A random set of genes from the scRNA-seq, with number comparable to the largest TF binding group (1000 genes) were used as control. (H) IGV track (*Robinson et al., 2011*) showing ChIP-peak overlap (red arrows) at common transcriptional target genes in cell mobility (*Sox9, Pdgfra, Snai1, Lamb1, Dlx2*) in NCC development and neurogenesis (*Jag1, Foxb1, Sox2, En1, Zic3, Dll3*) repressed at early migration. Gene diagrams are indicated (bottom row).

The online version of this article includes the following figure supplement(s) for figure 4:

**Figure supplement 1.** TWIST1 ChIP-seq experiment in ESC-derived neuroepithelial cells.
**Figure supplement 2.** Expression of key target genes identified in scRNA-seq analysis of E 8.5-E10.5 NCCs.
**Figure supplement 3.** Motif enrichment analysis of overlapping and unique ChIP-seq peaks between TWIST1 and CHD8.

(*Figure 4H*; *Supplementary file 4*). This integrative analysis revealed that the TWIST1-chromatin regulators shared genomic targets that are harbored in open chromatin in the NECs.

To pinpoint more specifically when TWIST1-chromatin regulators are required and better interpret their transcriptional activities in light of the in vivo dynamics and timing of target gene activity, we examined relevant gene activities in the E8.5- E10.5 mouse NCCs scRNA-seq datasets of NCCs traced by *Wnt1-Cre* and *Sox10-Cre* reporters (*Soldatov et al., 2019*). The first clue came from the expression of *Twist1, Chd7, Chd8,* and *Whsc1* in NCCs clusters that are ordered in developmental pseudotime: neural tube, delaminatory, early migratory, migratory 1, migratory 2, sensory, autonomic and mesenchyme (*Figure 4—figure supplement 2A–C*). *Twist1* displayed stage-specific dynamics and initially peaked in the early migratory NCC followed by exponential activation while progressing to the mesenchyme. On the other hand, the three interacting partners expressed rather ubiquitously throughout all NCC populations (*Figure 4—figure supplement 2C*).

We then examined the activities of genes with binding sites for TWIST1-chromatin regulators in their proximal regulatory elements. To narrow down to the most immediate targets, we limited the list to ChIP targets that are also responsive to *Twist1* conditional knockdown in the E9.5 NCCs (*Bildsoe et al., 2009*). Among all the NCC regulons, the binding sites of the TWIST1-module correlate best with the profile of early migratory NCCs (*Figure 4G*). We also noted that the marker genes of early migratory and neural tubes are mutually exclusive (*Figure 4—figure supplement 2D*). A substantial number of early migratory genes are repressed in the neural tube (62%), while the signature neural tube genes are downregulated at the early migratory stage (33%). TWIST1 and partners appeared to repress many of the neural tube or neurogenesis genes in the early migratory NCCs, including *Sox2, Foxb1, Jag1, En1, Zic3,* and *Dll3* (*Figure 4H*). In summary, in the in vivo context, the initial manifestation of Twist1-modular activity starts from delamination and correlates best with the early migratory stage. This early function may be important for the newly delaminated NCC progenitors to proceed to migratory stages, through the repression of neural tube signatures and the enhancement of cell migration and early NCC genes.

## TWIST1 is required for the recruitment of CHD8 to the regulatory region of target genes

To examine whether TWIST1 is necessary to recruit partner proteins to specific regions of co-regulated genes or vice versa, we examined chromatin binding of the endogenous proteins in NECs by ChIP-qPCR analysis (*Figure 5A*). As CHD8 correlates best with TWIST1 in their ChIP-seq profile surrounding TSS, we analyzed the pattern of recruitment of TWIST1 and CHD8 at the shared peaks near *Sox2, Epha3, Pdgfra,* and *Vegfa*. One of the peaks near the *Sox2* TSS demonstrated binding by

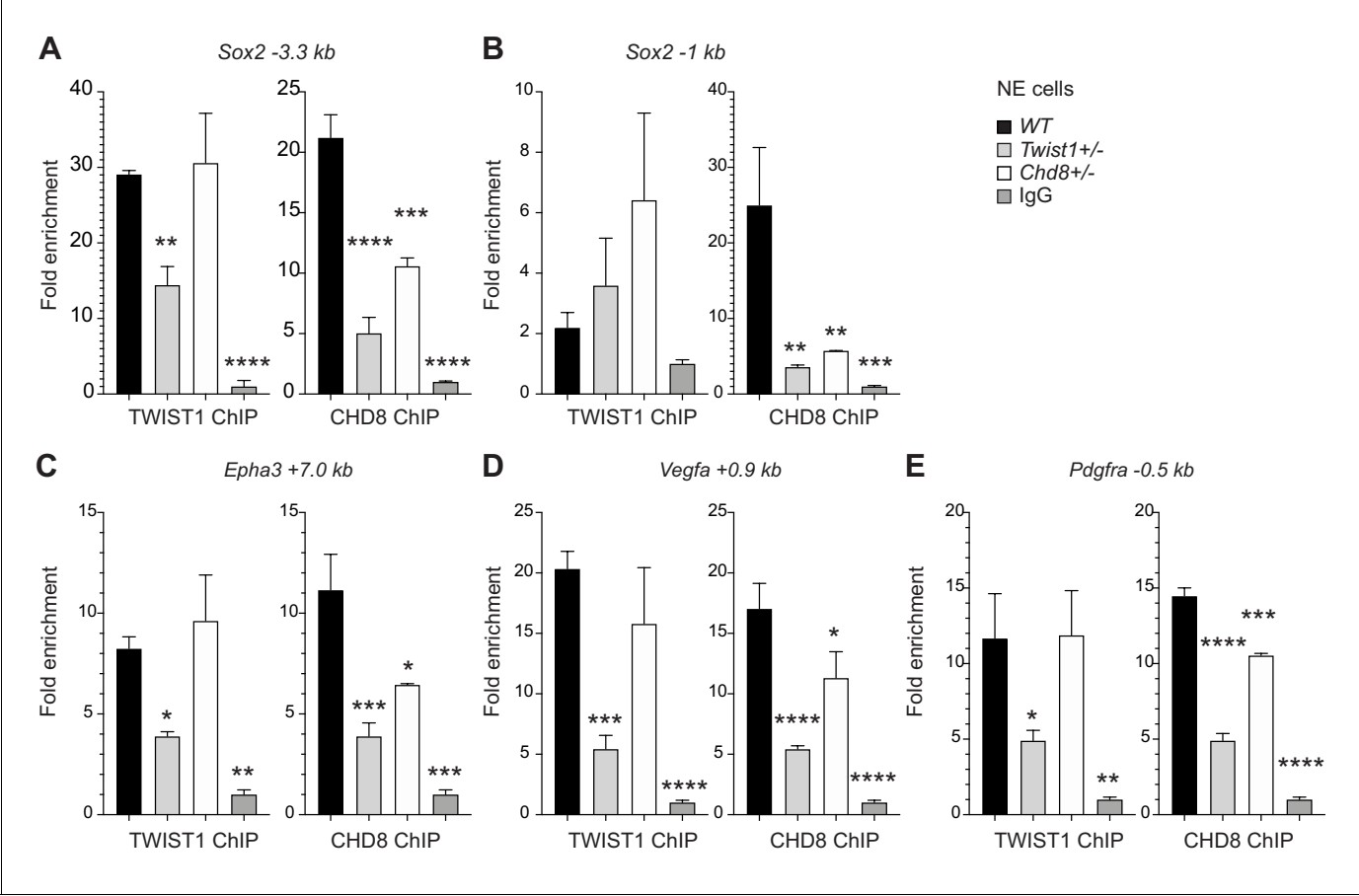

**Figure 5.** TWIST1 is required for the recruitment of CHD8 to the regulatory region of target genes. Binding of endogenous TWIST1 and CHD8 to overlapping genomic peak regions called by MACS2 (q < 0.05) were assessed by ChIP-qPCR. (**A–E**) qPCR quantification of genomic DNA from ChIP of endogenous TWIST1 or CHD8 proteins are shown as mean fold enrichment ± SE. ChIP experiments using anti-TWIST1 or anti-CHD8 antibodies against endogenous proteins were performed on *wildtype* (*WT*), *Twist1*[+/-] and *Chd8*[+/-] NECs derived from ESC (n = 3, day 3). qPCR results were normalized against signal from non-binding negative control region and displayed as fold change against IgG control. Each condition was compared against *WT* and P-values were generated using one-way ANOVA. *p<0.05, **p<0.01, ***p<0.001, ****p<0.0001. ns, not significant.

both TWIST1 and CHD8 (*Figure 5A,B*). In *Twist1*[+/-] or *Chd8*[+/-] NECs, the binding of TWIST1 or CHD8 at the peak was reduced. Interestingly, *Twist1*[+/-] mutation also diminished the binding of CHD8 yet *Chd8*[+/-] mutation did not affect TWIST1 binding (*Figure 5A*). For *Epha3*, *Vegfa*, and *Pdgfra*, peaks identified by ChIP-seq with H3K4me3 or H3K27ac modifications were tested. Partial loss of *Twist1* significantly reduced the recruitment of both TWIST1 and CHD8 but again, the loss of CHD8 only affected its own binding (*Figure 5C–E*). These findings support that TWIST1 binding is a prerequisite for the recruitment of CHD8.

## The TWIST1-chromatin regulators are necessary for cell migration and NCC ectomesenchyme propensity

As the TWIST1 and partners were found to regulate cell migration and BMP signaling pathways through target gene binding, we again took a loss-of-function approach and examined the synergic function of TWIST1-chromatin regulatory factors on cell motility in both NECs and NCCs. The emigration of NECs from the colonies was captured by time-lapse imaging and was quantified (see Materials and methods). While *Chd7*[+/-], *Chd8*[+/-], and *Whsc1*[+/-] mutant cells displayed marginally reduced motility, the motility of the *Twist1*[+/-] cells was compromised and further reduced in *Twist1*[+/-]; *Chd7*[+/-], *Twist1*[+/-]; *Chd8*[+/-], and *Twist1*[+/-]; *Whsc1*[+/-] compound mutant cells (*Figure 6A,B*). Additionally, to validate the functional interaction of these factors in the later phase of NCC development, and test whether they favor mesenchymal versus the alternative branches of NCC

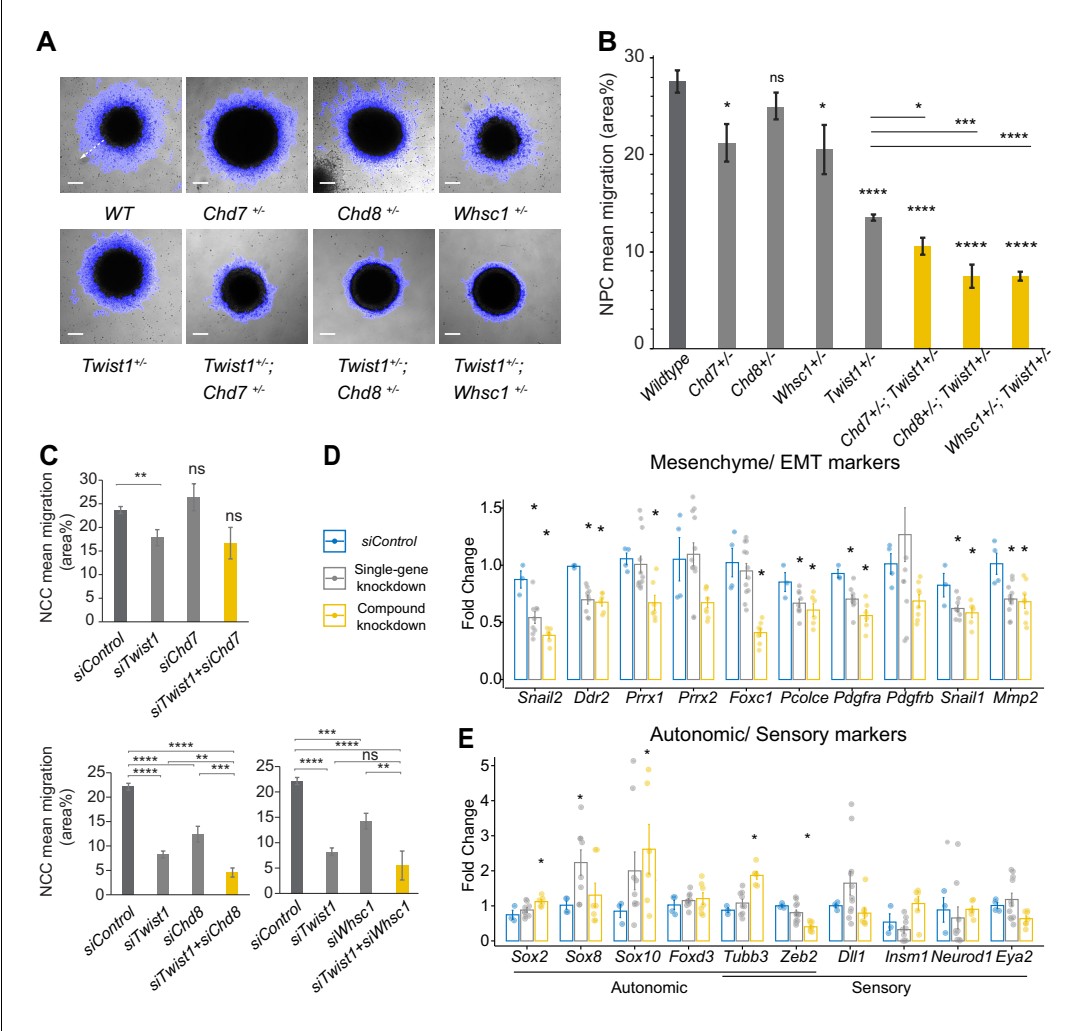

**Figure 6.** The TWIST1-chromatin regulators are necessary for cell migration and the NCC ectomesenchyme potential. (A) Dispersion of cells from the colony over 10 hr period in vitro (blue halo area). White arrow (shown in wildtype, *WT*) indicates the centrifugal cell movement. Bright-field time-lapse images were captured at set tile regions. Bar = 0.2 mm (B) Cell migration over 10 hr was quantified from time-lapse imaging data and plotted as mean area % ± SE for each cell type. n = 5 for each genotype. p-Values were computed by one-way ANOVA with Holm-sidak post-test. (C) Results of the scratch assay of O9-1 cells with siRNA knockdowns of *Twist1*, *Chd7*, *Chd8*, *Whsc1*, and control siRNA. Bright-field images were captured at set tile regions every 15 min over 10 hr. Cell migration was measured as mean area % traversed ± SE, in triplicate experiments for each genotype. Each condition was compared to *WT*. p-Values were computed by one-way ANOVA. *p<0.05, **p<0.01, ***p<0.001, ****p<0.0001. ns, not significant. (D, E) RT-qPCR quantification of expression of genes associated with EMT/ectomesenchyme, autonomic and sensory neuron fates, selected from E8.5-E10.5 mouse NCC scRNA-seq data. Gene expression is represented as fold change against control ± SE. The bar diagram shows the expression fold changes in cells treated with *siRNA* individually for *Twist/ Chd8* or *Whsc1* (pooled result of the treatments, gray bar) and *siRNA* for *Twist1* in combination with *Chd8* or *Whsc1* (pooled result of the treatments, yellow bar). Expression were normalized with the average expression of three housekeeping genes (*Gapdh, Tbp, Actb*). Each group was compared to control knockdown treatment. p-Values were computed by one-way ANOVA. *p<0.05.
The online version of this article includes the following figure supplement(s) for figure 6:

**Figure supplement 1.** Marker genes selected for qPCR analysis in O9-1 cells with TWIST1-CRM knockdown.

lineages, we performed knockdown analysis in O9-1 neural crest stem cells. O9-1 cells display the transcriptional signature of the mouse ectomesenchymal NCCs and could generate cranial mesen-chyme derivatives (osteoblasts, chondrocytes, smooth muscle cells) both in vitro and in vivo (*Ishii et al., 2012*). NCCs were treated with siRNA to knockdown *Chd7*, *Chd8*, and *Whsc1* activity individually (single-gene knockdown) and in combination with *Twist1*-siRNA (compound knockdown; *Figure 3—figure supplement 1C*, see Materials and methods). NCCs treated with *Chd8*-siRNA or *Whsc1*-siRNA but not *Chd7*-siRNA showed impaired motility (relative to control-siRNA treated cells),

which was exacerbated by the additional knockdown of *Twist1* (*Figure 6C*). We also performed qPCR on cell type markers highlighted in scRNA-seq analysis of E8.5 - E10.5 mouse NCCs (*Soldatov et al., 2019*; *Figure 6—figure supplement 1*). Specifically, we focused on the analysis of genes important for bifurcation events across the ectomesenchyme, autonomic, and sensory branches, as these genes may best report trans-differentiation activity (*Soldatov et al., 2019*). Among these genes, we prioritized those with DNA-binding sites for TWIST1 in their regulatory elements. Impaired motility in *Twist1, Chd8* and *Whsc1* knockdowns was accompanied by reduced expression of EMT genes (*Snail2, Ddr2, Pcolce, Pdgfrb, Mmp2*) and ectomesenchyme markers (*Prrx1, Prrx2, Foxc1, Snai1*; *Figure 6D*). Combined knockdowns had a stronger impact on the expression of the target genes than individual knockdowns for *Twist1, Chd8,* and *Whsc1* (*Figure 6D*). On the other hand, changes in the expression of non-mesenchymal genes, that is, sensory/ autonomic neuron markers, were less robust (*Figure 6E*). The significant upregulation of *Sox2, Sox8, Sox10,* and *Tubb3* may indicate a switch to autonomic neuron fate, which is the state immediately adjacent to the mesenchyme (*Soldatov et al., 2019*). Genes more specific for sensory neurons were either below detection or not significantly affected. Prolonged knockdown treatment may be required to detect changes of these cell fate markers. These findings suggested that the persistent activity of TWIST1-CHD8/WHSC1 is required to confer ectomesenchyme propensity (cell mobility, EMT, and mesenchyme differentiation), and repress neurogenic differentiation.

## TWIST1 and chromatin regulators for cell fate choice in neuroepithelial cells and lineage trajectory of neural crest cells

The genomic, transcriptomic and embryo phenotypic data collectively pointed to a requirement of TWIST1-chromatin regulators in the newly delaminated NCCs for progression towards early migratory state. To better understand how TWIST1-chromatin regulators coordinate NCC identities at early specification, we studied the role of the module factors during neural differentiation of ESCs in vitro. ESCs were cultured in neurogenic differentiation media, followed by selection and expansion of NECs (*Figure 7A*; *Bajpai et al., 2010*; *Varshney et al., 2017*). All cell lines progressed in the same developmental trajectory (*Figure 7B–i*) and generated Nestin-positive rosettes typical of NECs (*Figure 7D*). We assessed the lineage propensity of neuroepithelial cells derived from single-gene heterozygous ESCs (*Twist1*$^{+/-}$, *Chd7*$^{+/-}$, *Chd8*$^{+/-}$, and *Whsc1*$^{+/-}$) and compound heterozygous ESCs (*Twist1*$^{+/-}$;*Chd7*$^{+/-}$, *Twist1*$^{+/-}$;*Chd8*$^{+/-}$, and *Twist1*$^{+/-}$;*Whsc1*$^{+/-}$). Samples were collected at day 0 (ESCs), day 3 and day 12 of differentiation and assessed for the expression of cell markers and ChIP-seq target genes (*Supplementary file 5*). Genes were clustered into three groups by patterns of expression: activation, transient activation, and repression (*Figure 7B ii*, black, red, gray clusters). Notably, *Chd7, Chd8,* and *Whsc1* clustered with NCC specifiers that were activated transiently during differentiation (*Figure 7B ii*, red).

NCC and NSC marker genes responded inversely to the combined perturbation of *Twist1* and chromatin regulators. Compound loss-of-function (LOF) reduced expression of the NCC specifiers and unleashed the expression of NSC TFs in Day-12 NECs (*Figure 7C*, second and third row). In single-gene heterozygous cells, we observed only modest or no change in the gene expression. *Sox2*, a driver of the NSC lineage and a repressor of NCC formation (*Mandalos et al., 2014*), was repressed concurrently with the increased expression of *Twist1* and the chromatin regulators during neurogenic differentiation (*Figure 7C*). However, in the compound heterozygous cells, *Sox2* transcript and protein were both upregulated compared to the wild-type cells, together with NSC markers TUBB3 and NES (*Figure 7C–E*). Finally, the EMT genes were only affected by the compound knockdown at the ectomesenchyme stage (*Figure 7C*, bottom row; see also *Figure 6D,E*).

We focused on the effect of gene perturbation on the cell fate bias in late NECs by examining the expression of a broader panel of neural tube/NSC signatures (*Briscoe et al., 2000*; *Alaynick et al., 2011*; *Kutejova et al., 2016*; *Figure 4—figure supplement 1D*). The difference between WT and mutant cells in the dataset is primarily driven by changes in NCC specifiers and NSC TFs. In the compound mutant NECs, in addition to NCC identity (*Tfap2a, Msx1, Msx2, Zic1, Id1,* and *Id2*), expression of dorsal NSC markers were attenuated (*Gli3, Rgs2, Boc,* and *Ptn*; *Figure 7F,G*). Meanwhile, the pan- and ventral-NSC markers *Sox2, Pax6, Olig2, Foxa2,* and *Cited2* were ectopically induced (*Figure 7F,G*: genes in red). ChIP-seq data showed that TWIST1, CHD7 and CHD8 directly bind to the promoters of most of these genes (*Figure 4G*, S5A, *Supplementary file 4*).

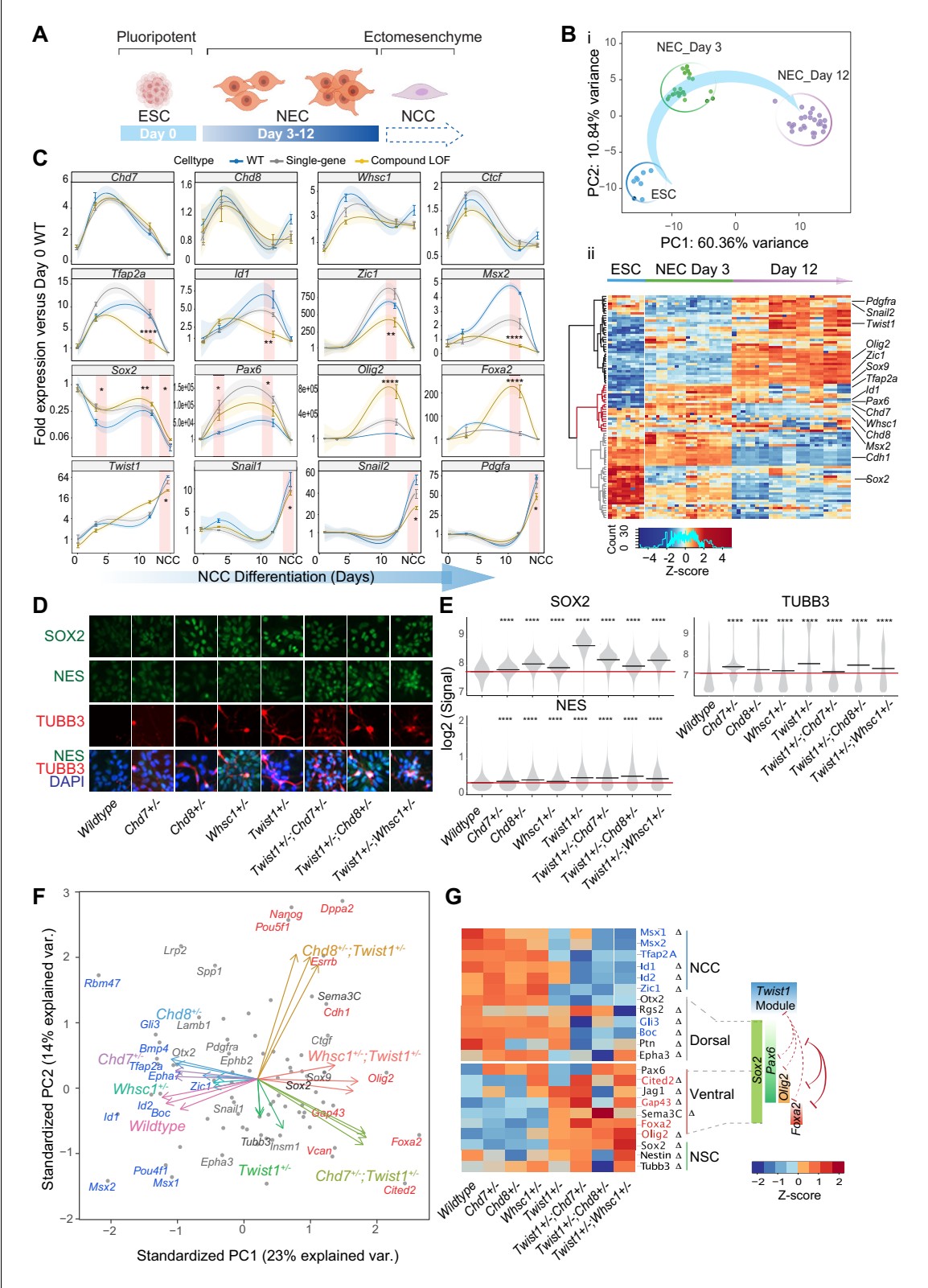

**Figure 7.** The TWIST1-chromatin regulators predispose NCC propensity and facilitate dorsal-ventral neuroepithelial specification. (**A**) Experimental strategy of neural differentiation in vitro (*Bajpai et al., 2010*; *Varshney et al., 2017*). (**B**) (i) Principal component analysis (PCA) of the Fluidigm high-throughput qPCR data for all cell lines collected as ESC, and neuroepithelial cells (NECs) at day 0, day 3, and day 12 of differentiation, respectively. Differentiation trajectory from ESC to NEC is shown for the first two PC axes. (ii) Heatmap clustering of normalized gene expressions for all cell lines:
*Figure 7 continued on next page*

Figure 7 continued

n = 3 for each genotype analyzed at day 3 and day 12 of neuroepithelium differentiation and n = 1 for ESCs. Clusters indicate activated (black), transiently activated (red) and repressed (gray) genes during neural differentiation. Z-score (color-coded) is calculated from $\log_2$ transformed normalized expression. (C) Profiles of expression of representative genes during neural differentiation (day 0 to NCC). Mean expression ± standard error (SE) are plotted for wildtype, single-gene heterozygous (average of $Twist1^{+/-}$, $Chd7^{+/-}$, $Chd8^{+/-}$, and $Whsc1^{+/-}$) or compound heterozygous (average of $Twist1^{+/-}$; $Chd7^{+/-}$, $Twist1^{+/-}$;$Chd8^{+/-}$, and $Twist1^{+/-}$;$Whsc1^{+/-}$) groups. For NCCs, samples were collected O9-1 cells with siRNA knockdown of single-gene or combinations of $Twist1$ and one of the partners. Gene expression were normalized against the mean expression value of three housekeeping genes ($Gapdh$, $Tbp$, $Actb$), and then the expression of day 0 wild-type ESCs. Shading of trend line represents 90% confidence interval. Red stripes indicate stages when target gene expressions were significantly affected by the double knockdown. -Values were calculated using one-way ANOVA. *p<0.05, **p<0.01, ***p<0.001, ****p<0.0001. ns, not significant. (D) Immunofluorescence of SOX2 and selected NSC markers and (E) quantification of signal intensity ± SE in single cells of indicated genotypes (X-axis). p-Values were generated using one-way ANOVA with Holm-sidak post-test. *p<0.05, **p<0.01, ***p<0.001, ****p<0.0001. (F) PCA plot of the NECs (day 12) showing genes with highest PC loadings (blue = top 10 loading, red = bottom 10 loading), and vector of each genotype indicating their weight on the PCs. (G) Heatmap of genes associated with neural tube to NCC transitions: NCC specifiers, dorsal-, ventral-, pan-NSC in mutant versus wild-type NECs. Progenitor identities along the neural tube, the reported master TFs and the co-repression (red solid line) of dorsal and ventral progenitors are illustrated on the right (*Briscoe et al., 2000*; *Alaynick et al., 2011*; *Kutejova et al., 2016*). Repression (red) or promotion (green) of cell fates by TWIST1-module based on the perturbation data are indicated in dashed lines. Genes with the highest PC loadings were indicated in same colors as in D. Z-scores (color-coded) were calculated from $\text{Log}_2$ fold-change against wildtype cells. Changes in gene expressions were significant (by one-way ANOVA). Genes identified as targets in at least two ChIP-seq datasets among TWIST1, CHD7, and CHD8 are labeled with Δ.

The online version of this article includes the following source data and figure supplement(s) for figure 7:

**Source data 1.** Normalized gene expression table from Fluidigm high-throughput qPCR analysis.
**Figure supplement 1.** Molecular model of the fate decision between neural crest cells (NCC) and neural stem cells (NSC).

Collectively, the findings implicated that in the NEC progenitor populations, TWIST1-chromatin regulators may promote the dorsal-most NCC propensity by counteracting SOX2 and other NSC TFs. Loss of function of the module leads to the reversion of the early NCC progenitors to neural-tube-like cells, which may underpin the severe deficiency of NCCs and loss of their derivatives observed in the mutant embryos (*Figure 3*).

## Discussion

### Proteomic screen and network-based inference of NCC epigenetic regulators

Analysis of protein-protein interaction is a powerful approach to identify the connectivity and the functional hierarchy of different genetic determinants associated with an established phenotype (*Song and Singh, 2009*; *Mitra et al., 2013*; *Sahni et al., 2015*; *Cowen et al., 2017*). We used TWIST1 as an anchor point and the BioID methodology to visualize the protein interactome necessary for NCCs development. Network propagation exploiting a similarity network built on prior associations enabled the extraction of clusters critical for neural crest function and pathology. Using this high-throughput analytic pipeline, we were able to identify the core components of the TWIST1-CRM that guides NCC lineage development.

Among the interacting factors were members of the chromatin regulation cluster, which show dynamic component switching between cell types, and may confer tissue-specific activities. The architecture of the modular network reflects the biological organization of chromatin remodeling machinery, which comprises multi-functional subunits with conserved and cell-type-specific components (*Meier and Brehm, 2014*). Previous network studies reported that disease-causal proteins exist mostly at the center of large clusters and have a high degree of connectivity (*Jonsson and Bates, 2006*; *Ideker and Sharan, 2008*). We did not observe an overall correlation between disease probability and the degree of connectivity or centrality for factors in the TWIST1 interactome (*Figure 1—figure supplement 2F*). However, the topological characteristics of the chromatin regulatory cluster resembled the features of disease modules and enriched for craniofacial phenotypes. In contrast, the 'ribosome biogenesis' module that was also densely inter-connected, was void of relevant phenotypic association (*Figure 1C*). Network propagation is, therefore, an efficient way to identify and prioritize important clusters while eliminating functionally irrelevant ones.

Based on these results, we selected core TWIST1-CRM epigenetic regulators CHD7, CHD8, and WHSC1, and demonstrated their physical and functional interaction with TWIST1. In the progenitors of the NCCs, these factors displayed overlapping genomic occupancy that correlated with the active chromatin marks in the fate specification genes in the neuroepithelium.

## Attribute of TWIST1 interacting partners in NCC development

Combinatorial perturbation of the disease 'hot-spots' in TWIST1-CRM impacted adversely on NCC specification and craniofacial morphogenesis in mouse embryos, which phenocopy a spectrum of human congenital malformations associated with NCC deficiencies (*Johnson et al., 1998*; *Chun et al., 2002*; *Cai et al., 2003*; *Bosman et al., 2005*; *Bernier et al., 2014*; *Schulz et al., 2014*; *Battaglia et al., 2015*; *Etchevers et al., 2019*). These observations revealed CHD8 and WHSC1 as putative determinants for NCC development and neurocristopathies. While CHD8 is associated with autism spectrum disorder (*Bernier et al., 2014*; *Katayama et al., 2016*), its function for neural crest development has never been reported. Here, we demonstrated that the loss of *Chd8* affected NCC migration and trigeminal sensory nerve formation in vivo, in a *Twist1*-dependent manner. We showed that TWIST1 occupancy is a requisite for CHD8 recruitment to common target genes. CHD8 may initiate chromatin opening and recruit H3-lysine tri-methyltransferases (*Zhao et al., 2015*) such as WHSC1 (*Figure 7—figure supplement 1*). The TWIST1-CHD8 complex may also repress neurogenic genes by blocking the binding of the competitive TFs. The specificity for NCC differentiation genes might be achieved when TWIST1, CHD8 and additional factors bind adjacently to each other, either sequentially or simultaneously. ChIP-seq peaks shared between TWIST1 and CHD8 or unique to each of them were enriched in different sets of motifs matched to various transcription factors (*Figure 4—figure supplement 3A*). Interestingly, only factors binding to TWIST1+CHD8 peaks show enriched expression in the delaminatory and migratory NCC populations (*Figure 4—figure supplement 3B*). These results suggested that TWIST1-CHD8 module may interact with additional factors, such as DLX1 and SOX10, to enhance NCC identity. We also showed that WHSC1 is required in combination with TWIST1 to promote NCC fate and tissue patterning. Unlike CHD8 and WHSC1, CHD7 has been previously implicated in neurocristopathy (CHARGE syndrome) and the motility of NCCs (*Schulz et al., 2014*; *Okuno et al., 2017*). Our study has corroborated these findings while also showing that CHD7 interacts with TWIST1 to promote NCC differentiation. One limitation of the mutant study is that there was no parallel comparison of the mutant phenotype in chimeras derived from multiple mutant cell lines to rule out the possibility of off-target gene-editing. However, in view of the potential variation of the ESC contribution to the chimera, We took a more productive approach to analyze multiple chimeric embryos from one line to glean a consistent phenotype that may inform the impact of genetic interaction Twist1 and the chromatin regulators on development.

In sum, we propose the TWIST1-CRM as a unifying model that connects previously unrelated regulatory factors implicated in different rare diseases and predicts their functional inter-dependency in NCC development (*Figure 7—figure supplement 1*). Other epigenetic regulators identified in the TWIST1-interactome, such as PBRM1, ZFP62, and MGA, may contribute to the activity of this regulatory module in the NCC lineage.

## The competition between TWIST1 module and SOX2 in cell fate decision

The segregation of neuroepithelial cells to NCC and NSC lineages is the first event of NCC differentiation. Our results show that the lineage allocation may be accomplished by the opposing activity of core members of the TWIST1-CRM and NSC TFs such as SOX2. *Sox2* expression is continuously repressed in the NCC lineage (*Wakamatsu et al., 2004*; *Cimadamore et al., 2011*; *Soldatov et al., 2019*), likely through direct binding and inhibition by TWIST1-CHD8 at *Sox2* promotor. In *Twist1*$^{+/-}$; *Chd8*$^{+/-}$ mutant embryos, the aberrant upregulation of *Sox2* correlated with deficiency of NCC derivatives and the expanded neuroepithelium of the embryonic brain. In a similar context, *Sox2* overexpression in chicken neuroepithelium blocks the production of TFAP2α-positive NCC (resulting in the loss of cranial nerve ganglia) and suppresses the expression of EMT genes and NCC migration (*Wakamatsu et al., 2004*; *Remboutsika et al., 2011*). On the contrary, conditional knockout of *Sox2*

results in ectopic production and abnormal migration of NCCs, and attenuation of the neuroepithelium (*Mandalos et al., 2014*).

## The phasic activity of TWIST1 and chromatin regulators in NCC differentiation

NCCs are derived from the neuroepithelium in a series of cell fate specification events (*Soldatov et al., 2019*). Classical studies of systemic and conditional *Twist1* knockout embryos have shed light on the in vivo implications of our molecular characterizations. Neural tube thickening and distortion were observed in the homozygous null mutant embryo. Cell number in the neural tube was doubled but was not due to altered cell proliferation or reduced cell death (*Vincentz et al., 2008*). *Twist1* null mutant has an expanded domain of *Wnt1-Cre* expression in the neural tube, and the NCC cells are frequently accumulated in the vicinity of the neuroepithelium (*Chen and Behringer, 1995*; *Soo et al., 2002*; *Vincentz et al., 2008*). Based on these phenotypes and our data from the knockout NECs, is possible that in the absence of *Twist1* activity, the process of cellular delamination is impaired leading to the retention of the NCC progenitors in the neuroepithelium or at ectopic sites.

TWIST1 and the chromatin regulators cooperatively drive the progression along the lineage trajectory at different phases of NCC differentiation. *Twist1* is activated from delamination and its expression steadily increases during differentiation. The functional interaction of TWIST1 and the chromatin regulators may commence at the transition from delaminatory to early migratory stage, coinciding with the first peak of *Twist1* expression. The expressions of NCC specification (*Msx1/2, Zic1*) or migratory genes (*Tfap2a*) were compromised by the loss of the TWIST1-chromatin regulators, albeit they are activated before *Twist1* during development (*Soldatov et al., 2019*). The early activity of *Twist1*, *Chd7*, *Chd8*, and *Whsc1* module might stabilize the activity of these early NCC genes. Loss of the module in NECs leads to reversion to NSC fate at the expense of the NCC lineage. In the post-migratory NCCs, modular activity of *Twist1* and *Chd8/ Whsc1* promotes the ectomesenchyme propensity while represses alternative cell fates such as autonomic neurons. *Chd7* activity was not connected with EMT in the NCCs, suggesting that its role may be different from CHD8 and WHSC1. The early versus late TWIST1-module activities are associated with the activation of different groups of target genes, suggesting that phasic deployment of the regulatory module may navigate the cells along the trajectory of NCC development.

Overexpression of *Twist1* in NCCs disrupts the formation of thoracic sympathetic chain ganglia, of the autonomic nervous system (*Vincentz et al., 2013*). On the other hand, in the *Wnt1-Cre* conditional *Twist1* knockout mutants, differentiation of cardiac neural crest cells into neuronal cells reminiscent of those in the sympathetic ganglia was found ectopically in the cardiac outflow tract (*Vincentz et al., 2013*). These aggregates of cardiac NCCs expressed pan-neuronal *Tubb3* and markers specific for the autonomic nerve cells (*Sox10, Phox2b*, and *Ascl1*), but did not express sensory neuron markers (*TrkA, Pou4f1*, and *NeuroD1*). Similarly, NCCs losing the function of TWIST1-module in vitro showed sign of mesenchyme to autonomic state trans-differentiation, whereas the sensory markers in these cells remain repressed. In the heterozygote null mutants, we did not observe any ectopic gain of neurons, but loss and disorganization of trigeminal nerves (composed of sensory and motor neurons), similar to the phenotype of null mutants (*Ota et al., 2004*). This may result from the loss of early function of *Twist1* in early migratory NCC formation, rather than its late activity on automimic-mesenchymal bifurcation.

In conclusion, by implementing an analytic pipeline to decipher the TWIST1 interactome, we have a glimpse of the global molecular hierarchy of NCC development. We have characterized the cooperative function of core components of TWIST1-CRM including the TWIST1 and chromatin regulators CHD7, CHD8, and WHSC1. We demonstrated that this module is a dynamic nexus to drive molecular mechanisms for orchestrating NCC lineage progression and repressing NSC fate, enabling the acquisition of ectomesenchyme propensity. The TWIST1-chromatin regulators and the NSC regulators therefore coordinate the molecular cross-talk between the ectomesenchyme and neurogenic progenitors of the central and peripheral nervous systems, which are often affected concurrently in a range of human congenital diseases.

# Materials and methods

## Key resources table

| Reagent type (species) or resource | Designation | Source or reference | Identifiers | Additional information |
|---|---|---|---|---|
| Aantibody (mouse monoclonal) | α-TWIST | Abcam | cat# ab50887; RRID:AB_883294 | 1:1000 |
| Antibody (rabbit polyclonal) | α-CHD7 | Abcam | cat# ab117522; RRID:AB_10938324 | 1:5000 |
| Antibody (rabbit polyclonal) | α-CHD8 | Abcam | cat# ab114126; RRID:AB_10859797 | 1:10,000 |
| Antibody (mouse monoclonal) | α-WHSC1 | Abcam | cat# ab75359; RRID:AB_1310816 | 1:5000 |
| Antibody (rabbit polyclonal) | α-SOX2 | Abcam | cat# ab59776; RRID:AB_945584 | 1:1000 |
| Antibody (mouse monoclonal) | α-VINCULIN | Sigma | cat# V9131; RRID:AB_477629 | 1:1000 |
| Antibody (mouse monoclonal) | α-OCT3/4 | Santa Cruz | cat# sc-5279; RRID:AB_628051 | 1:1000 |
| Antibody (rabbit polyclonal) | α-TFAP2A | Abcam | cat# ab52222; RRID:AB_867683 | 1:1000 |
| Antibody (mouse monoclonal) | α-NESTIN | Abcam | cat# ab7659; RRID:AB_2298388 | 1:1000 |
| Antibody (mouse monoclonal) | α-FLAG | Sigma | cat# F1804; RRID:AB_262044 | 1:1000 |
| Antibody (mouse monoclonal) | α-neurofilament | DSHB | cat#2H3; | 1:1000 |
| Antibody (rabbit polyclonal) | α-HA | Abcam | cat# ab9110; RRID:AB_307019 | 1:1000 |
| Transfected construct (M. musculus) | siChd7 #1 | Sigma-Aldrich | SASI_Mm02_00298181 | |
| Transfected construct (M. musculus) | siChd7 #2 | Sigma-Aldrich | SASI_Mm01_00234434 | |
| Transfected construct (M. musculus) | siChd7 #3 | Sigma-Aldrich | SASI_Mm01_00234436 | |
| Transfected construct (M. musculus) | siChd8 #1 | Sigma-Aldrich | SASI_Mm02_00351688 | |
| Transfected construct (M. musculus) | siChd8 #2 | Sigma-Aldrich | SASI_Mm02_00351689 | |
| Transfected construct (M. musculus) | siChd8 #3 | Sigma-Aldrich | SASI_Mm02_00351691 | |
| Transfected construct (M. musculus) | siWhsc1 #1 | Sigma-Aldrich | SASI_Mm01_00278608 | |
| Transfected construct (M. musculus) | siWhsc1 #2 | Sigma-Aldrich | SASI_Mm01_00278610 | |
| Transfected construct (M. musculus) | siWhsc1 #3 | Sigma-Aldrich | SASI_Mm02_00295201 | |
| Transfected construct (M. musculus) | siTwist1 | Sigma-Aldrich | SASI_Mm01_00043025 | |
| Transfected construct (M. musculus) | siRNA Universal Negative Control #1 | Sigma-Aldrich | SIC001 | |
| Transfected construct (M. musculus) | siRNA Universal Negative Control #2 | Sigma-Aldrich | SIC002 | |
| Cell line (M. musculus) | O9-1 neural crest stem cell | Millipore | Millipore | |

*Continued on next page*

*Continued*

| Reagent type (species) or resource | Designation | Source or reference | Identifiers | Additional information |
|---|---|---|---|---|
| Cell line (*M. musculus*) | 3T3 | ATCC | ATCC | |
| Cell line (*M. musculus*) | A2loxCre ESCs | Kyba Lab | | Lillehei Heart Institute, Minnesota, USA |
| Cell line (*M. musculus*) | A2loxCre Twist1$^{+/-}$ | This paper | | See Materials and methods |
| Cell line (*M. musculus*) | A2loxCre Twist1$^{-/-}$ | This paper | | See Materials and methods |
| Cell line (*M. musculus*) | A2loxCre Twist1$^{+/-}$; Chd7$^{+/-}$ | This paper | | See Materials and methods |
| Cell line (*M. musculus*) | A2loxCre Twist1$^{+/-}$; Chd8$^{+/-}$ | This paper | | See Materials and methods |
| Cell line (*M. musculus*) | A2loxCre Twist1$^{+/-}$; Whsc1$^{+/-}$ | This paper | | See Materials and methods |
| Cell line (*M. musculus*) | A2loxCre Chd7$^{+/-}$ | This paper | | See Materials and methods |
| Cell line (*M. musculus*) | A2loxCre Chd8$^{+/-}$ | This paper | | See Materials and methods |
| Cell line (*M. musculus*) | A2loxCre Whsc1$^{+/-}$ | This paper | | See Materials and methods |

## Cell culture and BioID protein proximity-labeling

O9-1 cells were purchased from Millipore and 3T3 cells were purchased from ATCC. A2loxCre mouse ESCs were a gift from the Kyba Lab (Lillehei Heart Institute, Minnesota, USA). Derivatives of A2loxCre ESCs were generated in the lab. Cell line identities were authenticated by genotyping, and all cell lines were tested free of mycoplasma. O9-1 cells (passage 20–22, Millipore cat. #SCC049) were maintained in O9-1 medium: high glucose DMEM (Gibco), 12.5% (v/v) heat-inactivated FBS (Fisher Biotec), 10 mM β-mercaptoethanol, 1X non-essential amino acids (100X, Thermo Fisher Scientific), 1% (v/v) nucleosides (100X, Merck) and 10 mil U/mL ESGRO mouse leukaemia inhibitory factor (Merck) and 25 ng/mL FGF-2 (Millipore, Cat. #GF003). For each replicate experiment, $1.5 \times 10^6$ cells per flask were seeded onto 4*T75 flasks 24 hr before transfection. The next day *PcDNA 3.1/ Twist1-BirA\*-HA* plasmid or *PcDNA 3.1/ GFP-BirA\*-HA* plasmid was transfected into cells using Lipofectamine 3000 (Life Technologies) according to the manufacturer's instructions. Biotin (Thermo Scientific, cat. #B20656) was applied to the medium at 50 nM. Cells were harvested 16 hr post-transfection, followed by snap-freeze liquid nitrogen storage or resuspension in lysis buffer. All steps were carried out at 4°C unless indicated otherwise. Cells were sonicated on the Bioruptor Plus (Diagenode), 30 s on/off for five cycles at high power. An equal volume of cold 50 mM Tris-HCl, pH 7.4, was added to each tube, followed by two 30 s on/off cycles of sonication. Lysates were centrifuged for 15 min at 14,000 rpm. Protein concentrations were determined by Direct Detect Infrared Spectrometer (Merck).

Cleared lysate with equal protein concentration for each treatment was incubated with pre-blocked streptavidin Dynabeads (MyOne Streptavidin C1, Invitrogen, cat. #65002) for 4 hr. Beads were collected and washed sequentially in Wash Buffer 1–3 with 8 min rotation each, followed by quick washes with cold 1 mL 50 mM Tris·HCl, pH 7.4, and 500 μL triethylammonium bicarbonate (75 mM). Beads were then collected by spinning (5 min at 2000 × *g*) and processed for mass spectrometry analysis.

### Lysis buffer

500 mM NaCl, 50 mM Tris-HCl, pH 7.5, 0.2% SDS 0.5% Triton.
Add 1x Complete protease inhibitor (Roche), 1 mM DTT fresh.

### Wash buffer 1
2% SDS.

### Wash buffer 2
0.1% sodium deoxycholate, 1% Triton, 1 mM EDTA, 1 mL, 500 mM NaCl, 50 mM HEPES-KOH, pH 7.5.

### Wash buffer 3
0.1% sodium deoxycholate, 0.5% NP40 (Igepal), 1 mM EDTA, 250 mM NaCl, 10 mM Tris-HCl, pH 7.5.

## Liquid chromatography with tandem mass spectrometry

Tryptic digestion of bead-bound protein was performed in 5% w/w trypsin (Promega, cat. #V5280), 50 mM triethylammonium bicarbonate buffer at 37°C overnight. The supernatant was collected and acidified with trifluoroacetic acid (TFA, final concentration 0.5% v/v). Proteolytic peptides were desalted using Oligo R3 reversed phase resin (Thermo Fisher Scientific) in stage tips made in-house (*Rappsilber et al., 2007*). Peptides were fractioned by hydrophilic interaction liquid chromatography using an UltiMate 3000 HPLC (Thermo Fisher Scientific) and a TSKgel Amide-80 HILIC 1 mm ×250 mm column. Peptides were eluted in a gradient from 100% mobile phase B (90% acetonitrile, 0.1% TFA, 9.9% water) to 60% mobile phase A (0.1% TFA, 99.9% water) for 35 min at 50 µL/min and fractions collected in a 96-well plate, followed by vacuum centrifugation to dryness. Dried peptide pools were reconstituted in 0.1% formic acid in the water, and 1/10th of samples were analyzed by LC-MS/MS.

Mass spectrometry was performed using an LTQ Velos-Orbitrap MS (Thermo Fisher Scientific) coupled with an UltiMate RSLCnano-LC system (Thermo Fisher Scientific). A volume of 5 µL was loaded onto a 5 mm C18 trap column (Acclaim PepMap 100, 5 µm particles, 300 µm inside diameter, Thermo Fisher Scientific) at 20 µL/ min for 2.5 min in 99% phase A (0.1% formic acid in water) and 1% phase B (0.1% formic acid, 9.99% water and 90% acetonitrile). The peptides were eluted through a 75 µm inside diameter column with integrated laser-pulled spray tip packed to a length of 20 cm with Reprosil 120 Pur-C18 AQ 3 µm particles (Dr. Maisch). The gradient was from 7% phase B to 30% phase B in 46.5 min, to 45% phase B in 5 min, and to 99% phase B in 2 min. The mass spectrometer was used to apply 2.3 kV to the spray tip via a pre-column liquid junction. During each cycle of data-dependent MS detection, the ten most intense ions within m/z 300–1500 above 5000 counts in a 120,000 resolution orbitrap MS scan were selected for fragmentation and detection in an ion trap MS/MS scan. Other MS settings were: MS target was 1,000,000 counts for a maximum of 500 ms; MS/MS target was 50,000 counts for a maximum of 300 ms; isolation width, 2.0 units; normalized collision energy, 35; activation time 10 ms; charge state one was rejected; mono-isotopic precursor selection was enabled; dynamic exclusion was for 10 s.

## Proteomic data analysis

### Pre-processing of raw mass spectrometry data

Raw MS data files were processed using Proteome Discoverer v.1.3 (Thermo Fisher Scientific). Processed files were searched against the UniProt mouse database (downloaded Nov 2016) using the Mascot search engine version 2.3.0. Searches were done with tryptic specificity allowing up to two missed cleavages and tolerance on mass measurement of 10 ppm in MS mode and 0.3 Da for MS/MS ions. Variable modifications allowed were acetyl (Protein N-terminus), oxidized methionine, glutamine to pyro-glutamic acid, and deamidation of asparagine and glutamine residues. Carbamidomethyl of cysteines was a fixed modification. Using a reversed decoy database, a false discovery rate (FDR) threshold of 1% was used. The lists of protein groups were filtered for first hits.

Processing and analysis of raw peptide-spectrum match (PSM) values were performed in R following the published protocol (*Waardenberg, 2017*). Data were normalized by the sum of PSM for each sample (*Figure 1—figure supplement 2B*), based on the assumption that the same amount of starting materials was loaded onto the mass spectrometer for the test and control samples. A PSM value of 0 was assigned to missing values for peptide absent from the sample or below detection level (*Sharma et al., 2009*). Data points filtered by the quality criterion that peptides had to be

present in at least two replicate experiments with a PSM value above 2. The normalized and filtered dataset was fitted under the negative binomial generalized linear model and subjected to the likelihood ratio test for TWIST1 vs. control interactions, using the msmsTest and EdgeR packages (*Robinson et al., 2010*; *Gregori et al., 2019*). Three biological replicates each from O9-1 and 3T3 cells were analyzed. One set of C3H10T1/2 cell line was analyzed. A sample dispersion estimate was applied to all datasets. Stringent TWIST1-specific interactions in the three cell lines were determined based on a threshold of multi-test adjusted p-values (adjp) <0.05 and fold-change >3.

## Network propagation for functional identification and novel disease gene annotation

Prior knowledge of mouse protein functional associations, weighted based on known protein-protein interaction (PPI), co-expression, evolutionary conservation, and test mining results, were retrieved by the Search Tool for the Retrieval of Interacting Genes (STRING) (*Szklarczyk et al., 2015*). Intermediate confidence (combined score) of >0.4 was used as the cut-off for interactions. The inferred network was imported into Cytoscape for visualization (*Shannon et al., 2003*). We used MCL algorism (*Enright et al., 2002*), which emulates random walks between TWIST1 interacting proteins to detect clusters in the network, using the STRING association matrix as the probability flow matrix. Gene Ontology and transcriptional-binding site enrichment analysis for proteins were obtained from the ToppGene database (*Chen et al., 2009*), with a false-discovery rate <0.05. The enriched functional term of known nodes was used to annotate network neighbors within the cluster with unclear roles.

Heat diffusion was performed on the network, using 22 genes associated with human and mouse facial malformation (HP:0001999, MP:0000428) as seeds. A diffusion score of 1 was assigned to the seeds, and these scores were allowed to propagate to network neighbors, and heat stored in nodes after set time = 0.25 was calculated. NetworkAnalyzer (*Assenov et al., 2008*), which is a feature of Cytoscape, was used to calculate nodes' Degree (number of edges), Average Shortest Path (connecting nodes), and Closeness Centrality (a measure of how fast information spreads to other nodes).

## Co-immunoprecipitation

### Protein immunoprecipitation

For the analysis of protein localization, transfection was performed using Lipofectamine 3000 (Life Tech) according to manufacturer instructions with the following combinations of plasmids: pCMV-*Twist1-FLAG* plus one of (*pCMV-gfp-HA, pCMV-Tcf3-HA, pCMV-Prrx1-HA, pCMV-Prrx2-HA, pCMV-Chd7-HA, pCMV-Chd8-HA, pCMV-Dvl1-HA, pCMV-Smarce1-HA, pCMV-Tfe3-HA, pCMV-Whsc1-HA, pCMV-Hmg20a-HA)*. The cell pellet was lysed and centrifuged at 14,000 x *g* for 15 min. Cleared lysate was incubated with α-TWIST1/ α-FLAG antibody (1 µg/mL) at 4°C for 2 hr with rotation. Protein-G agarose beads (Roche) were then added, and the sample rotated for 30 min at RT °C. Beads were washed in ice-cold wash buffer six times and transferred to new before elution in 2x LDS loading buffer at 70°C for 10 min. Half the eluate was loaded on SDS-PAGE with the 'input' controls for western blot analysis.

### Western blotting

Protein was extracted using RIPA buffer lysis (1× PBS, 1.5% Triton X-100, 1% IGEPAL, 0.5% Sodium Deoxycholate, 0.1% SDS, 1 mM DTT, 1x Complete protease inhibitor [Roche]) for 30 min at 4°C under rotation. The lysate was cleared by centrifugation at 15000 g, and protein concentration was determined using the Direct Detect spectrometer (Millipore). 20 µg of protein per sample was denatured at 70°C for 10 min in 1× SDS Loading Dye (100 mM Tris pH 6.8, 10% (w/v) SDS, 50% (w/v) Glycerol, 25%(v/v) 2-Mercaptoethanol, Bromophenol blue) and loaded on a NuPage 4–12% Bis-Tris Gel (Life Technologies, Cat. #NP0322BOX). Electrophoresis and membrane transfer was performed using the Novex (Invitrogen) system following manufacturer instructions.

Primary antibodies used were mouse monoclonal α-TWIST1 (1:1000, Abcam, Cat. #ab50887), mouse monoclonal [29D1] α-WHSC1/NSD2 (1:5000, Abcam, Cat. #ab75359), rabbit polyclonal α-CHD7 (1:5000, Abcam, Cat. #ab117522), rabbit polyclonal α-CHD8 (1:10000, Abcam, Cat. #ab114126), mouse α-α-tubulin (1:1000, Sigma, Cat. #T6199), rabbit α-HA (1:1000, Abcam, Cat. #ab9110) and mouse α-FLAG M2 (Sigma, Cat. #F1804). Secondary antibodies used were HRP-

conjugated donkey α-Rabbit IgG (1:8000, Jackson Immunoresearch, Cat. #711-035-152) and HRP-conjugated donkey α-Mouse IgG (1:8000, Jackson Immunoresearch, Cat. #711-035-150).

## GST pull-down

### Production and purification of recombinant proteins

Prokaryotic expression plasmids pGEX2T with the following inserts GST-*Twist1*, GST-*N'Twist1*, GST-*C'Twist1*, GST-*Twist1bhlh*, GST-*Twist1TA, or* GST were transfected in BL21 (DE3) *Escherichia coli* bacteria (Bioline). Bacterial starter culture was made by inoculation of 4 mL Luria broth with 10 μg/mL ampicillin, and grown 37°C, 200 rpm overnight. Starter culture was used to inoculate 200 mL Luria broth media with 10 μg/mL ampicillin and grown at 37°C, 200 rpm until the optical density measured OD600 was around 0.5–1.0. The culture was cooled down to 25°C for 30 min before Iso-propyl β-D-1-thiogalactopyranoside (IPTG) was added to the media at a final concentration of 1 mM. Bacteria were collected by centrifugation 4 hr later at 8000 rpm for 10 min at 4°C.

Bacteria were resuspended in 5 % volume of lysis buffer (10 mM Tris-Cl, pH 8.0; 300 mM NaCl; 1 mM EDTA, 300 mM NaCl, 10 mM Tris.HCl [pH 8.0], 1 mM EDTA, 1x Complete protease inhibitor [Roche], 1 mM PMSF, 100 ng/mL leupeptin, 5 mM DTT) and nucleus were released by 3 rounds of freeze/thaw cycles between liquid nitrogen and cold water. The sample was sonicated for 15 s x 2 (consistent; intensity 2), with 3 min rest on ice between cycles. Triton X-100 was added to a final concentration of 1%. The lysate was rotated for 30 min at 4°C and centrifuged at 14,000 rpm for 15 min at 4°C.

The supernatant was collected and rotated with 800 μL of 50% Glutathione Sepharose 4B slurry (GE, cat. # 17-0756-01) for 1 hr, at 4°C. Beads were then loaded on MicroSpin columns (GE cat. #27-3565-01). Column was washed three times with wash buffer (PBS 2X, Triton X-100 0.1%, imidazole 50 mM, NaCl 500 mM, DTT 1 mM, 1x Complete protease inhibitor [Roche]) before storage in 50% glycerol (0.01% Triton). Quantity and purity of the recombinant protein on beads were assessed by SDS polyacrylamide gel electrophoresis (SDS-PAGE, NuPAGE 4–12% bisacrylamide gel, Novex) followed by Coomassie staining or western blot analysis with anti-TWIST1 (1:1000), anti-GST (1:1000) antibody. Aliquots were kept at −20°C for up to 6 months.

### GST pulldown

Cell pellet ($5 \times 10^6$) expressing HA-tagged TWIST1 interaction candidates were thawed in 300 μL hypotonic lysis buffer (HEPES 20 mM, MgCl2 1 mM, Glycerol 10%, Triton 0.5%. DTT 1 mM, 1x Complete protease inhibitor [Roche], Benzo nuclease 0.5 μl/mL) and incubated at room temperature for 15 min (for nuclease activity). An equal volume of hypertonic lysis buffer (HEPES 20 mM, $NaCl_2$500 mM, $MgCl_2$1 mM, Glycerol 10%, DTT 1 mM, 1x Complete protease inhibitor [Roche]) was then added to the lysate. Cells are further broken down by passaging through gauge 25 needles for 10 strokes and rotated at 4°C for 30 min. After centrifugation at 12,000 x g, 10 min, 200 μL lysate was incubated with 10 μL bead slurry (or the same amount of GST fusion protein for each construct decided by above Coomassie staining). Bait protein capture was done at 4°C for 4 hr with rotation.

Beads were collected by spin at 2 min at $800 \times g$, 4°C, and most of the supernatant was carefully removed without disturbing the bead bed. Beads were resuspended in 250 μL ice-cold wash buffer, rotated for 10 min at 4°C and transferred to MicroSpin columns that were equilibrated with wash buffer beforehand. Wash buffer was removed from the column by spin 30 s at $100 \times g$, 4°C. Beads were washed for four more times quickly with ice-cold wash buffer before eluting proteins in 2X LDS loading buffer 30 μL at 70°C, 10 min, and characterized by western blotting.

## Generation of mutant ESC by CRISPR-Cas9 editing

CRISPR-Cas9-edited mESCs were generated as described previously (*Sibbritt et al., 2019*). Briefly, 1–2 gRNAs for target genes were ligated into pSpCas9(BB)−2A-GFP (PX458, addgene plasmid #48138, *a gift from Feng Zhang*). Three μg of pX458 containing the gRNA was electroporated into 1 × 10⁶ A2loxCre ESCs or A2loxCre Twist1+/- cells (clone T2-3, generated by the Vector and Genome Engineering Facility at the Children's Medical Research Institute) using the Neon Transfection System (Thermo Fisher Scientific). Electroporated cells were plated as single cells onto pre-seeded lawns of mouse embryonic fibroblasts (MEF), and GFP expressing clones grown from single cells were selected under the fluorescent microscope. In total, 30–40 clones were picked for each

electroporation. For mutant ESC genotyping, clones were expanded and grown on a gelatin-coated plate for three passages, to remove residue MEFs contamination.

For genotyping, genomic lysate of ESCs was used as input for PCR reaction that amplified region surrounding the mutation site (± 200–500 bp flanking each side of the mutation). The PCR product was gel purified and sub-cloned into the pGEM-T Easy Vector System (Promega) as per manufacturer's protocol. At least 10 plasmids from each cell line were sequenced to ascertain monoallelic frameshift mutation and exclude biallelic mutations.

## Generation of mouse chimeras from ESCs

ARC/s and *DsRed.T3* mice were purchased from the Australian Animal Resources Centre and maintained as homozygous breeding pairs. ESC clones with monoallelic frameshift mutations and the parental *A2LoxCre* ESC line were used to generate chimeras. Embryo injections were performed as previously described (*Sibbritt et al., 2019*). Briefly, 8–10 ESCs were injected per eight-cell *DsRed.T3* embryo (harvested at 2.5 dpc from super-ovulated *ARC/s* females crossed to *DsRed.T3* stud males) and incubated overnight. Ten to 12 injected blastocysts were transferred to each E2.5 pseudo-pregnant ARC/s female recipient. E9.5 and E11.5 embryos were collected 6 and 8 days after transfer to pseudo-pregnant mice. Embryos showing red fluorescent signal indicating no or low ESC contribution were excluded from the phenotypic analysis. Animal experimentations were performed in compliance with animal ethics and welfare guidelines stipulated by the Children's Medical Research Institute/Children's Hospital at Westmead Animal Ethics Committee, protocol number C230.

## Whole-mount fluorescent immunostaining of mouse embryos

Whole-mount fluorescent immunostaining of mouse embryos was performed by following the procedure of *Adameyko et al., 2012* with minor modifications. Embryos were fixed for 6 hr in 4% paraformaldehyde (PFA) and dehydrated through a methanol gradient (25%, 50%, 75%, 100%). After 24 hr of incubation in 100% methanol at 4°C, embryos were transferred into bleaching solution (1 part of 30% hydrogen peroxide to 2 parts of 100% methanol) for another 24 hr (4°C). Embryos were then washed with 100% methanol (10 min x3 at room temperature), post-fixed with Dent's Fixative (dimethyl sulfoxide: methanol = 1:4) overnight at 4°C.

Embryos were blocked for 1 hr on ice in blocking solution (0.2% BSA, 20% DMSO in PBS) with 0.4% Triton. Primary antibodies mouse 2H3 (for neurofilament 1:1000) and rabbit $\alpha$-TFAP2A (1:1000) or were diluted in blocking solution and incubated for four days at room temperature, and secondary antibodies (Goat $\alpha$-Rabbit Alexa Fluor 633; Goat $\alpha$-Mouse Alexa Fluor 488 and DAPI, Thermo Fisher Scientific) were incubated overnight in blocking solution at room temperature. Additional information of the antibodies used are listed in Key Resources Table. Embryos were cleared using BABB (1part benzyl alcohol: two parts benzyl benzoate), after dehydration in methanol, and imaged using a Carl Zeiss Cell Observer SD spinning disc microscope. Confocal stacks through the embryo were acquired and then collapsed. Confocal stacks were produced containing ~150 optical slices. Bitplane IMARIS software was used for 3D visualization and analysis of confocal stacks. Optical sections of the 3D embryo were recorded using ortho/oblique functions in IMARIS software. The surface rendering wizard tool was used to quantify SOX2 expression in the ventricular zone by measuring the immunofluorescence intensity on three separate z-plane sections per volume of the region of each embryo. The data were presented graphically as the ratio of intensity/ volume.

## Generation of TWIST1 inducible expression ESC line

ESC lines generated are listed in Key Resources Table. A2loxCre Mouse ESCs (*Mazzoni et al., 2011*) was a gift from Kyba Lab (Lillehei Heart Institute, Minnesota, USA). A2loxCre with Twist1 bi-allelic knockout background was generated by CRISPR-Cas9, as described below. The inducible *Twist1* ESC line was generated using the inducible cassette exchange method described previously (*Iacovino et al., 2014*). The TWIST1 coding sequence was then cloned from the mouse embryo cDNA library into the p2lox plasmid downstream of the Flag tag (*Iacovino et al., 2014*). The plasmid was transfected into A2loxCre (*Twist1 $^{-/-}$*) treated with 1 µg/mL doxycycline for 24 hr. The selection was performed in 300 µg/mL of G418 (Gibco) antibiotic for 1 week. Colonies were then picked and tested for TWIST1 expression following doxycycline treatment.

## NEC differentiation of the ESCs

ESC lines generated in this study were differentiated into neural epithelial cells (NECs) following established protocols (*Bajpai et al., 2010*; *Varshney et al., 2017*) with minor modifications. ESCs were expanded in 2i/LIF media (*Ying et al., 2008*) for 2–3 passages. Neurogenic differentiation was initiated by plating ESC in AggreWells ($1 \times 10^6$ per well) using feeder independent mESC. Colonies were then lifted from AggreWells and grown in suspension in Neurogenic Differentiation Media supplemented with 15% FBS with gentle shaking for 3 days. Cell colonies were transferred to gelatin-coated tissue culture plates and cultured for 24 hr at 37°C under 5% $CO_2$.

Cells were selected in insulin-transferrin-selenium (ITS)-Fibronectin media for 6–8 days at 37°C and 5% $CO_2$, with a change of media every other day. Accutase (Stemcell Technologies) was used to dissociate cells from the plate, allowing the removal of cell clumps. NECs were collected by centrifugation and plated on Poly-L-ornithine (50 µg/mL, Sigma-Aldrich) and Laminin (1 µg/mL, Novus Biological) coated dishes. For expansion of the cell line, cells were cultured in Neural Expansion Media (1.5 mg/mL Glucose, 73 µg/mL L-glutamine, 1x N2 media supplement [R and D systems] in Knockout DMEM/F12 [Invitrogen], 10 ng/mL FGF-2, and 1 µg/mL Laminin [Novus Biologicals]). During this period, cells were lifted using Accutase and cell rosette clusters were let settle and were removed for two passages to enrich for pre-EMT NCC populations.

## Chromatin immunoprecipitation sequencing (ChIP-seq)

ESC with genotype *Twist1$^{-/-}$; Flag-Twist1* O/E and *Twist1$^{-/-}$* were differentiated into NEC for 3 days following established protocol (*Varshney et al., 2017*) and were collected in ice-cold DPBS. Following a cell count, approximately $2 \times 10^7$ cells were allocated per cell line per ChIP. ChIP-seq assays were performed as previously described (*Bildsoe et al., 2016*). In brief, chromatin was crosslinked and sonicated on the Bioruptor Plus (Diagenode) using the following program: 30 s on/off for 40 min on High power. The supernatant was incubated with α-TWIST1 (Abcam, at. #ab50887) antibody conjugated Dynabeads overnight at 4°C. The protein-chromatin crosslinking is reversed by incubation at 65°C for 6 hr. The DNA is purified using RNase A and proteinase K treatments, extracted using phenol-chloroform-isoamyl alcohol (25:24:1, v/v) and precipitated using glycogen and sodium acetate. The precipitated or input chromatin DNA was purified and converted to barcoded libraries using the TruSeq ChIP Sample Prep Kit (Illumina). Then 101 bp paired-end sequencing was performed on the HiSeq 4000 (Illumina).

## ChIP-sequencing data analysis

ChIP-seq quality control results and analysis can be found in *Figure 4—figure supplement 1*. Adaptors from raw sequencing data were removed using Trimmomatic (*Bolger et al., 2014*) and aligned to the *mm10* mouse genome (GENCODE GRCm38.p5); (*Frankish et al., 2019*) using BWA aligner (*Li and Durbin, 2009*), and duplicates/unpaired sequences were removed using the picardtools (http://broadinstitute.github.io/picard/). MACS2 package (*Zhang et al., 2008*) was used for ChIP-seq peak calling for both *Twist1$^{-/-}$; Flag-Twist1 O/E* and *Twist1$^{-/-}$* IP samples against genomic input. IDR analysis was performed using the p-value as the ranking measure, with an IDR cut-off of 0.05. Peak coordinates from the two replicates were merged, using the most extreme start and end positions. The raw and processed data were deposited into the NCBI GEO database and can be accessed with the accession number GSE130251.

## ChIP-seq integrative analysis

Public ChIP-seq datasets for CHD7, CHD8, and histone modifications in NECs were selected based on the quality analysis from the Cistrome Data Browser (http://cistrome.org/db/#/) and ENCODE guideline (*ENCODE Project Consortium, 2012*; *Mei et al., 2017*). Datasets imported for analysis are listed in *Supplementary file 6*. To facilitate comparison with datasets generated from human samples, TWIST1 ChIP sequences were aligned to the hg38 human genome by BWA. ChIP peak coordinates from this study were statistically compared using fisher's exact test (cut-off: p-value<0.05, odds ration >10) and visualized using Jaccard similarity score. Analysis were performed with BEDTools (*Quinlan and Hall, 2010*). ChIP-seq peaks for TWIST1, CHD7 and CHD8 were extended to uniform 1 kb regions, and regions bound by single factors or co-occupied by two or three factors were identified. The Genomic Regions Enrichment of Annotations Tool (GREAT) was

used to assigns biological functions to genomic regions by analyzing the annotations of the nearby genes (*McLean et al., 2010*). Significance by both binomial and hypergeometric test (p<0.05) were used as cut-off. Genes with TSS ± 5 kb of the peaks were annotated using ChIPpeakAnno package in R. List of target genes was compared between each CHD7, CHD8, and TWIST1. Bam files for each experiment were converted to bigwig files for ChIP-seq density profile, chromosome footprint, and IGV track visual analysis.

## scRNA-seq and DNA binding site enrichment analysis

scRNA-seq datasets for cranial E8.5, vagal/trunk E9.5, hindlimb/tail E10.5 and cardiac E10.5 Wnt1-traced, E9.5 anterior and E9.5 posterior Sox10-traced NCCs were obtained from GEO database (GSE129114) (*Soldatov et al., 2019*). Tables of per-gene read counts in each cell were imported into R. Single-cell datasets were pre-processed using Seurat package (*Stuart et al., 2019*), which includes pre-processing, normalization, and joint analysis of multiple datasets. Only the cells with more than 4000 expressed genes were included in the downstream analysis. Additionally, only genes with more than 10 mapped reads and detected in at least 10 cells were considered in the downstream analysis. For reproducible result, we imported cell clustering, annotation and t-SNE embedding for visualization from the original publication (http://pklab.med.harvard.edu/ruslan/neural.crest.html). Marker genes enriched in each cluster compared to all other clusters were determined using the Wilcoxon Rank Sum test.

Enrichment TWIST1-module transcriptional targets in NCC cluster markers genes were analysed using goseq R package (*Young et al., 2010*). To narrow down to the most immediate targets, we limited the list to ChIP targets that are also responsive to *Twist1* conditional knockdown in the E9.5 NCCs (*Bildsoe et al., 2009*). Random sampling was performed to generate a null distribution for each motif category and calculate its significance for over-representation amongst NCC regulons.

## O9-1 siRNA treatment and scratch assay

Scratch Assays were performed on O9-1 cells following transient siRNA lipofectamine transfections. O9-1 cells were seeded at a density of $0.5 \times 10^5$ cells per well on Matrigel-coated 24-well plates on the day of transfection. 20 pmol of siRNA for candidate gene (*Chd7, Chd8*, or *Whsc1*) and 20 pmol siRNA for *Twist1* or control was applied per well (24-well-plate), plus 3 μL lipofectamine RNAiMAX reagent (Thermo Fisher Scientific, cat. #13778075), following manufacturer protocol. Knockdown efficiency was assessed by qRT-PCR (*Figure 7—figure supplement 1*).

Forty-eight hr after transfection, a scratch was made in the confluent cell monolayer. Live images were taken with the Cell Observer Widefield microscope (ZEISS international) under standard cell culture conditions (37°C, 5% $CO_2$). Bright-field images were captured at set tile regions every 15 min over a 10-hr period. The total migration area from the start of imaging to when the first cell line closed the gap was quantified by Fiji software (*Schindelin et al., 2012*).

## cDNA synthesis, pre-amplification, and Fluidigm high-throughput RT-qPCR analysis

cDNA synthesis, from 1 μg total RNA from each sample, was performed using the RT2 Microfluidics qPCR Reagent System (Qiagen, Cat. # 330431). cDNAs were pre-amplified using the primer Mix for reporter gene sets (*Supplementary file 5*). High-throughput gene expression analysis (BioMarkTM HD System, Fluidigm) was then performed using the above primer set.

Raw data were extracted using the Fluidigm Real-Time PCR Analysis Software, and subsequent analysis was performed in R-studio. Ct values flagged as undetermined or higher than the threshold (Ct >24) were assigned as missing values. Samples with a measurement for only one housekeeping gene or samples with measurements for <30 genes were excluded from further analysis. Genes missing values for more than 30 samples were also excluded from further analysis. Data were normalized using expressions of the average of three housekeeping genes (*Gapdh, Tbp, Actb*). Regularized-log transformation of the count matrix was then performed, and the PCA loading gene was generated using functions in the DEseq2 package. Differential gene expression analysis was performed using one-way ANOVA.

## Acknowledgements

Our work was supported by the National Health and Medical Research Council (NHMRC) of Australia (Grant ID 1066832), the Australian Research Council (Grant DP 1094008) and Mr. James Fairfax (Bridgestar Pty Ltd). Imaging analysis was performed at the ACRF Telomere Analysis Centre and proteomics analysis was performed at the Biomedical Proteomics Facility, both supported by the Australian Cancer Research Foundation. XCF was supported by the University of Sydney International Postgraduate Research Scholarship, the Australian Postgraduate Award and the CMRI Scholarship; KEK was supported by The Danish Council for Independent Research and FP7 Marie Curie Actions – COFUND (DFF – 1325–00154) and the Carlsberg Foundation (CF15-1056 and CF16-0066); PO is the CMRI Norman Gregg Research Fellow; MEG was supported by NHMRC (grant ID 1079160); NF was supported by a University of Sydney Post-Doctoral Fellowship and the CMRI Norman Gregg Research Fellowship and PPLT is an NHMRC Senior Principal Research Fellow (Grant ID 1003100, 1110751).

## Additional information

### Funding

| Funder | Grant reference number | Author |
| --- | --- | --- |
| National Health and Medical Research Council | 1066832 | Mark E Graham<br>Patrick PL Tam |
| Australian Research Council | 1094008 | Xiaochen Fan<br>Patrick PL Tam |
| University of Sydney | | Xiaochen Fan |
| Children's Medical Research Institute | | Xiaochen Fan<br>Pierre Osteil<br>Nicolas Fossat |
| Carlsbergfondet | CF15-1056 | Kasper Engholm-Keller |
| National Health and Medical Research Council | 1079160 | Mark E Graham<br>Patrick PL Tam |
| National Health and Medical Research Council | 1003100 | Mark E Graham<br>Patrick PL Tam |
| National Health and Medical Research Council | 1110751 | Mark E Graham<br>Patrick PL Tam |
| Carlsbergfondet | CF16-0066 | Kasper Engholm-Keller |
| Marie Curie Cancer Care | DFF – 1325–00154 | Kasper Engholm-Keller |

The funders had no role in study design, data collection and interpretation, or the decision to submit the work for publication.

### Author contributions

Xiaochen Fan, Conceptualization, Formal analysis, Validation, Investigation, Visualization, Methodology, Writing - original draft, Writing - review and editing; V Pragathi Masamsetti, Validation, Visualization, Writing - review and editing; Jane QJ Sun, Validation; Kasper Engholm-Keller, Investigation, Methodology, Writing - review and editing; Pierre Osteil, Methodology, Writing - review and editing; Joshua Studdert, Resources, Methodology; Mark E Graham, Resources, Supervision, Funding acquisition, Methodology, Writing - review and editing; Nicolas Fossat, Supervision, Funding acquisition, Methodology, Writing - review and editing; Patrick PL Tam, Conceptualization, Resources, Supervision, Funding acquisition, Writing - original draft, Writing - review and editing

### Author ORCIDs

Xiaochen Fan (iD) https://orcid.org/0000-0002-4316-0616
Mark E Graham (iD) http://orcid.org/0000-0002-7290-1217
Patrick PL Tam (iD) https://orcid.org/0000-0001-6950-8388

## Ethics

Animal experimentation: Animal experimentations were performed in compliance with animal ethics and welfare guidelines stipulated by the Children's Medical Research Institute/Children's Hospital at Westmead Animal Ethics Committee, protocol number C230.

## Decision letter and Author response

Decision letter https://doi.org/10.7554/eLife.62873.sa1
Author response https://doi.org/10.7554/eLife.62873.sa2

# Additional files

## Supplementary files

- Supplementary file 1. BioID EdgeR test result table.
- Supplementary file 2. TWIST1 protein interaction module and Gene Ontology analysis.
- Supplementary file 3. Information on BioID candidates selected for validation. Cell line of origin of the candidate is listed. Expression data of the embryonic head was from published study (*Fan et al., 2016*). PSM: peptide sequence matches. Log2 FC = log2 transformed PSM fold-change between TWIST1-BirA*HA and GFP transfected O9-1 cells. Adjusted p-value was computed from dataset from O9-1 cell line, generated by the likelihood ratio test corrected by the Benjamini and Hochberg method in EdgeR (*Robinson et al., 2010*). Diffusion Rank: The rank of candidates in heat diffusion from genes associated with human and mouse facial malformation.
- Supplementary file 4. Integrative analysis of ChIP datasets.
- Supplementary file 5. BioMark reporter card setup.
- Supplementary file 6. Summary of external ChIP-seq datasets analyzed in this study.
- Transparent reporting form

## Data availability

All data generated or analyzed during this study are included in the manuscript and supporting files. Sequencing data have been deposited in GEO under accession code GSE130251. External data analyzed has been listed in Supplementary File 6.

The following dataset was generated:

| Author(s) | Year | Dataset title | Dataset URL | Database and Identifier |
|---|---|---|---|---|
| Fan X | 2020 | TWIST1 direct targets during embryonic stem cell differentiation [ChIP-seq] | https://www.ncbi.nlm.nih.gov/geo/query/acc.cgi?acc=GSE130251 | NCBI Gene Expression Omnibus, GSE130251 |

The following previously published datasets were used:

| Author(s) | Year | Dataset title | Dataset URL | Database and Identifier |
|---|---|---|---|---|
| Sugathan A, Biagioli M, Golzio C, Erdin S, Blumenthal I, Manavalan P, Ragavendran A, Brand H, Lucente D, Miles J, Sheridan SD, Stortchevoi A, Haggarty SJ, Katsanis N, Gusella JF, Talkowski ME | 2014 | CHD8 regulates neurodevelopmental pathways associated with autism spectrum disorder in neural progenitors [ChIP-Seq] | https://www.ncbi.nlm.nih.gov/geo/query/acc.cgi?acc=GSE61487 | NCBI Gene Expression Omnibus, GSE61487 |
| Ziller MJ, Edri R, | 2015 | Dissecting neural differentiation | https://www.ncbi.nlm. | NCBI Gene |

| | | | | |
|---|---|---|---|---|
| Yaffe Y, Donaghey J, Pop R, Mallard W, Issner R, Gifford CA, Goren A, Xing J, Gu H, Cachiarelli D, Tsankov A, Epstein C, Rinn JR, Mikkelsen TS, Kohlbacher O, Gnirke A, Bernstein BE, Elkabetz Y, Meissner A | | regulatory networks through epigenetic footprinting | nih.gov/geo/query/acc.cgi?acc=GSE62193 | Expression Omnibus, GSE62193 |
| Chai M, Sanosaka T, Okuno H, Zhou Z, Koya I, Banno S, Andoh-Noda T, Tabata Y, Shimamura R, Hayashi T, Ebisawa M, Sasagawa Y, Nikaido I, Okano H, Kohyama J | 2018 | AF22_H3K36me3 | https://www.ncbi.nlm.nih.gov/geo/query/acc.cgi?acc=GSM2902410 | NCBI Gene Expression Omnibus, GSM2902410 |
| Du Y, Liu Z, Cao X, Chen X, Chen Z, Zhang X, Jiang C | 2017 | Genome-wide maps of chromatin state during the differentiation of hESC into hNECs (ChIP-Seq) | https://www.ncbi.nlm.nih.gov/geo/query/acc.cgi?acc=GSM1973975 | NCBI Gene Expression Omnibus, GSM1973975 |
| Hikichi T, Matoba R, Ikeda T, Watanabe A, Yamamoto T, Yoshitake S, Tamura-Nakano M, Kimura T, Kamon M, Shimura M, Kawakami K, Okuda A, Okochi H, Inoue T, Suzuki A, Masui S | 2013 | Transcription factors interfering with dedifferentiation induce direct conversion | https://www.ncbi.nlm.nih.gov/geo/query/acc.cgi?acc=GSM1012189 | NCBI Gene Expression Omnibus, GSM1012189 |
| Mistri TK, Devasia AG, Chu LT, Ng WP, Halbritter F, Colby D, Martynoga B, Tomlinson SR, Chambers I, Robson P, Wohland T | 2015 | Selective influence of Sox2 on POU transcription factor binding in embryonic and neural stem cells | https://www.ncbi.nlm.nih.gov/geo/query/acc.cgi?acc=GSM1711445 | NCBI Gene Expression Omnibus, GSM1711445 |
| Kutejova E, Sasai N, Shah A, Gouti M, Briscoe J | 2016 | Neural progenitors adopt specific identities by directly repressing all alternative progenitor transcriptional programs | https://www.omicsdi.org/dataset/omics_ena_project/PRJEB7682 | The European Nucleotide Archive, ERS580651 |
| Soldatov R, Kaucka M, Kastriti ME, Petersen J, Chontorotzea T, Englmaier L, Akkuratova N, Yang Y, Häring M, Dyachuk V, Bock C, Farlik M, Piacentino ML, Boismoreau F, Hilscher MM, Yokota C, Qian X, Nilsson M, Bronner ME, Croci L, Hsiao W-YY, Guertin DA, Brunet J-FF, Consalez GG, Ernfors P, Fried K, Kharchenko PV, Adameyko I | 2019 | Spatio-temporal structure of cell fate decisions in murine neural crest | https://www.ncbi.nlm.nih.gov/geo/query/acc.cgi?acc=GSE129114 | NCBI Gene Expression Omnibus, GSE129114 |

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
