## [Decision Letter]

**Acceptance summary:**

This study uncovers mechanism underlying transcriptional control of neural crest development by identifying Twist binding partners and how they function together in vitro and in vivo. The study is well done and a valuable contribution to the literature.

**Decision letter after peer review:**

Thank you for submitting your article "TWIST1 and chromatin regulatory proteins interact to guide neural crest cell differentiation" for consideration by *eLife*. Your article has been reviewed by three peer reviewers, and the evaluation has been overseen by a Marianne Bronner as the Senior and Reviewing Editor. The following individuals involved in review of your submission have agreed to reveal their identity: Igor Adameyko (Reviewer #1); Anthony Firulli (Reviewer #2).

The reviewers have discussed the reviews with one another and the Reviewing Editor has drafted this decision to help you prepare a revised submission.

Summary:

In this well done manuscript, the authors examine the bHLH transcription factor TWIST1 and its interacting proteins in neural crest cell development using an unbiased screen. Given the important role of neural crest cells in craniofacial and cardiac developmental defects, the data are both useful and important.

Essential revisions:

The major problem is the claim that the regulation reported here is important for neural crest specification / induction. This cannot be the case, as Twist 1 starts to be expressed in mouse only during the delamination step according to published single cell data. The premigratory *Zic/Msx* positive neural crest shows no expression of *Twist1* before EMT markers kick in. The authors need to deal with this. It would be important to show in vivo expression data analysis and bring the conclusions in line with the timing in neural crest development.

In addition to these issues, the reviewers raise many smaller points which would be good to address. For this reason, I have included the full reviews below.

Reviewer #1:

This excellent study is focused on the mechanisms of action of *Twist1* in the neural crest cells and on the identification of core components of *Twist1* network. The authors performed an in-depth experimental study and sophisticated analysis to identify *Chd7/8* as the key partners of *Twist1* during NCC development. This identification and corresponding predictions later appeared consistent with experimental in vivo data including single and combinatorial gene knockout mouse models with phenotypes in the cranial neural crest. Overall, this study is important for the field.

However, I disagree with some secondary interpretations the authors give to their results. At the same time, the major conclusions stay solid. Below I discuss the most critical points.

1) *Chd7*, *Chd8* and *Whsc1* are ubiquitously expressed. Thus, the specificity of regulation is achieved via interactions with other, more cell type- and stage-specific, factors. This would be good to mention.

2) The authors suggest: "The phenotypic data so far indicate that the combined activity of TWIST1-chromatin regulators might be required for the establishment of NCC identity. To examine whether TWIST1- chromatin regulators are required for NCC specification from the neuroepithelium and to pinpoint its primary molecular function in early neural differentiation, we performed an integrative analysis of ChIP-seq datasets of the candidates".

This is a strange assumption, given that *Twist1* is expressed only staring from the NCC delamination stage in mouse cranial neural crest (Soldatov et al., 2019). It does not seem to correlate with premigratory NCC identity and the situation inside of the neural tube. The authors conclude: "Therefore, combinatorial binding sites for TWIST1, CHD7 and CHD8 may confer specificity for regulation of patterning genes in the NECs." Or, alternatively, they may confer the control of mesenchymal phenotype, downstream migration and fate biasing etc. I do not think the authors have good arguments to bring up induction or patterning of NCCs at the level of neural tube.

I have a good suggestion for the authors: I would extract the regulons from Soldatov et al. single cell data and run the binding site proximity check for the individual genes belonging to the gene modules /regulons specific to delamination and early NCC migration stages. I am curious, if the proximity of binding sites of *Twist1*-related crowd would rather correlate with genes from these specific regulons as compared to randomly selected regulons from the entire published single cell dataset.

Randomization/bootstrapping analysis are welcomed. So far, being an excellent study, this paper does not solve a problem of downstream (of *Twist1*) gene expression program in the neural crest cells. At the same time, this is what the author can try to obtain with their DNA binding data in combination with published single cell data. Repression of *Sox2* and upregulation of *Pdgfra* (reported in Figure 4) might be a part of this downstream program being in line with the published single cell gene expression data (Soldatov et al., 2019).

The authors conclude the paragraph: "Therefore, combinatorial binding sites for TWIST1, CHD7 and CHD8 may confer specificity for regulation of patterning genes in the NECs". Again, this is not a good or plausible explanation based on specificity of expression of suggested pattering genes (or visualized genes are poorly selected). Additionally, although I believe the obtained results are important and of a good quality, I would not call them "developmentally equivalent to ectomesenchymal NCCs" or other NCCs. This is because the in vitro system will never reflect the embryonic in vivo development with high accuracy (especially when it comes to patterning and positional identity). This might explain that some prominent binding positions and interpretations the authors give do not correspond the gene expression logic during neural crest development. Besides, *Twist1* and *Chd7/8* are naturally expressed in many other cell types and might target non-NCC genes (Vegfa?). This does not reduce the value of the data, but it is good to mention for the community.

3) Figure 2: *Twist1^-/+^Chd8^-/+^* is repeated two times in panel B (but the embryos look differently), although the authors most likely meant to show *Twist1^-/+^ Chd7^-/+^* in the second case. If this is indeed the case, the authors should also show a phenotype of *Chd7* KO.

4) The authors write: "Impaired motility in *Twist1*, *Chd8* and *Whsc1* knockdowns was accompanied by reduced expression of EMT genes (*Pdgfrα*, *Pcolce*, *Tcf12*, *Ddr2*, *Lamb1* and *Snai2*) (Figure 6D, Figure 3—figure supplement 1D) and ectomesenchyme markers (*Sox9*, *Spp1*, *Gli3*, *Klf4*, *Snai1*), while 375 genes that are enriched in the sensory neurons located in the dorsal root ganglia (Ishii et al., 2012) were upregulated (*Sox2*, *Sox10*, *Cdh1*, *Gap43*; Figure 6E).

– From the list of genes characterizing EMT, I can agree only on *Pdgfra* and *Snai2*, the rest is unspecific for EMT, and appears rather ubiquitous or specific to different cell populations (non-EMT).

– From the list of suggested ectomesenchyme markers, I cannot pick any gene that would be a bit specific for ectomesenchyme (within neural crest lineage) except for *Snai1*. *Sox9* is broadly expressed also in the trunk neural crest, *Spp1* and *Klf4* are not expressed in early mouse ectomesenchyme, *Gli3* is too broad and non-selective. I suggest to select other gene sets (check the expression with online PAGODA app from Soldatov et al):

http://pklab.med.harvard.edu/cgi-bin/R/rook/nc.p63-66.85-87.dbc.nc/index.html

– The choice of DRG genes is also non-optimal, as *Sox10* is pan-NCC, *Sox2* is expressed in early migrating crest and satellite glial cells of DRG and Schwann cell precursors, Gap43 and Cdh1 are not specific enough. these gene clearly suggest the beginning of neuro-glial fates or trunk neural crest bias. To be more precise and for claiming sensory neurons, the authors should come up with pro-neuronal genes such as neurogenins, *NeuroD, Isl1, Pou4f1, Ntrk* and many others.

Still, overall, I agree with author's main conclusions.

5) The authors write: "The genomic and embryo phenotypic data collectively suggest a requirement of TWIST1- chromatin regulators in the establishment of NCC identity in heterogeneous neuroepithelial 403 populations". Again, I do not think the authors can claim anything related to the establishment of NCC identity. NCC identity, in broad sense, includes NCC induction within the neural tube, at both trunk and cranial levels. In mice, *Twist1* is not expressed in trunk NCCs at all. At a cranial level, *Twist1* is expressed too late to be a NCC inducing or patterning gene. As I mentioned earlier, it comes up during delamination.

6) Figure 7G only partly corresponds to the positioning of the NCC markers in a mouse embryo. Id1 and Id2 are broadly expressed throughout all phases of NCC development and in entire dorsal neural tube beyond the NC region. Mentioning *Otx2* as a NCC specifier is strange. At the same time, *Msx1*, *Msx2*, *Zic1* are excellent genes! *Tfap2* is a bit too late, but still ok. Please keep in mind, *Msx1/2*, *Zic1* are expressed before *Twist1*, and, thus, *Twist1* can be downstream of this gene expression program. Also, these genes become downregulated quite soon upon delamination, whereas *Twist1*/*Chd7/8* expression stays (in vivo). Expression pattern of *Tfap2a* better corresponds to *Twist1*, although *Tfap2a* comes a bit before *Twist1*, and, besides, *Tfap2a* is expressed independently of *Twist1* in trunk NCC. Despite such gene expression divergence, *Twist1*-based network might provide positive feedback loops stabilizing the expression of other transcriptional programs that were originally induced by other factors. It might be good mentioning this to the readers. This "stabilizing role" of *Twist1* network can be a really important one. Given the incremental and combinatorial nature of the phenotype in vivo – this is most likely the case. I believe, these points are important to reflect in the Discussion section.

Reviewer #2:

The manuscript by Fan et. al is a comprehensive look into the bHLH protein TWIST1 and its interacting proteins in neural crest cell differentiation. The study employs an unbiased screen where a TWIST1-BirA fusion is used in conjunction with biotin linking to collect Twist protein transcriptional complexes. (BioID-Proximity-labeling, TWIST1-CRMs). The work appears carefully done and the data and impact of this study are high given the nature of NCCs being involved as key players in craniofacial and cardiac developmental defects. The association of TWIST1 with the chromatin helicases CHD7 and 8 is important to understand as numerous TWIST1 loss-of-function studies indicate that its roll in NCCs clearly is required for normal NCC function.

The NCC cell line O9-1 is used to collect the data. Genetic interactions between TW1, *Chd7*, *Chd8* and *Whsc1* are tested in genome edited ESCs. Overall, this is a well-executed, interesting and important study.

Reviewer #3:

Using BioID, the authors identified more than 140 proteins that potentially interact with transcription factor *Twist1* in a neural crest cell line. Most of these 140 *Twist1*-interactomes do not overlap with the 56 known *Twist1* binding partners during neural crest cell development (see below). By focusing on several strong *Twist1* binding partner candidates (particularly a novel candidate CHD8), the authors found:

1) *Twist1* interacts with these proteins via its N-terminal protein domain as demonstrated by co-IP.

2) Compound heterozygous mutation of *Chd8*, *Chd7* or *Whsc* and *Twist1* displayed more severe phenotype compared to heterozygous mutation of *Twist1* alone, for example, more significant reduction of the cranial nerve bundle thickness.

3) ChIPseq analysis of *Twist1* and CHD8 and key histone modifications revealed that the binding of *Chd8* strongly correlate with those of *Twist1*, to active enhancers that are also labeled by H3K4me3 and H3K27ac.

4) The binding of CHD8 requires the binding of *Twist1*, but not vice versa,

5) *Twist1*-*Chd8* regulatory module repress neuronal differentiation, and promotes neural crest cell migration, and potentially their differentiation into the non-neuronal cell types.

The authors use an impressive array of different techniques, both in vitro and in vivo, and yield consistent results. The manuscript is nicely written. The findings are nuanced, but the major conclusions are largely expected.

1) As the title states, the three key TWIST interacting factors that most of the study focuses on are chromatin regulators. However, the consequence of mutating these factors at the epigenetic level was not directly addressed, including the level of active histone modification, the accessibility of the *Twist1*/CDH co-bound promoters/enhancers, and the position of nucleosomes.

2) CRISPR-generated ESCs and chimera technology were used effectively to generate mutants. In comparison, the analysis of the phenotypes was rather cursory and can benefit from more in-depth molecular analysis. Especially, the altered genes found in mutant NEC and NCC in the last section of the study should be validated in mutants.

3) Across the manuscript, there were jumps from NCC to NEC and back. It will be important to justify why a certain cell type is selected for each analysis, focusing on the biological question at hand.

4) Using BioID, the authors detected 140 different proteins that interact with *Twist1*. However, only 4 of them overlap with the 56 known *Twist1* partners (Figure 1A). This result suggests that BioID identified almost a distinct set of *Twist1*-interacting proteins, compared to the published results. The authors need to discuss the discrepancy, and the underlying reasons.

5) The authors show that *Twist1* colocalize with *Cdh8*, and is required for the binding of *Cdh8*, thus suggest that *Twist1*-*Cdh8* form a regulatory module. Given the degenerate nature of bHLH factor binding motifs, it is likely that the binding of *Twist1*, and subsequently the binding of *Cdh8*, are dictated by other transcription factors. Therefore, a motif enrichment analysis should be done among the *Twist1*/*Cdh8* co-binding sites, and compare those motifs enriched in *Twist1*-only and *Cdh8*-only binding sites.

6) The increasing expression of DRG neurons genes in *Twist1*/*Cdh8* mutants suggests a possible transition from cranial NC to trunk NC. Therefore, the authors should examine the expression of marker genes accordingly.

---

## [Author Response]

Essential revisions:The major problem is the claim that the regulation reported here is important for neural crest specification / induction. This cannot be the case, as Twist 1 starts to be expressed in mouse only during the delamination step according to published single cell data. The premigratory Zic/Msx positive neural crest shows no expression of Twist1 before EMT markers kick in. The authors need to deal with this. It would be important to show in vivo expression data analysis and bring the conclusions in line with the timing in neural crest development.

We have examined the temporal profile of *Twist1* expression by mining the scRNA-seq data (Soldatov et al., 2019) and found that *Twist1* starts to be expressed in the delaminating cells. We agreed with the critiques and have removed the inference that TWIST1 involves in the specification/induction of neural crest cells.

In addition to these issues, the reviewers raise many smaller points which would be good to address. For this reason, I have included the full reviews below.

We have attended to the concerns raised in the review to our best ability. Please see the point-by-point response and the revision made to the manuscript.

Reviewer #1:This excellent study is focused on the mechanisms of action of Twist1 in the neural crest cells and on the identification of core components of Twist1 network. The authors performed an in-depth experimental study and sophisticated analysis to identify Chd7/8 as the key partners of Twist 1 during NCC development. This identification and corresponding predictions later appeared consistent with experimental in vivo data including single and combinatorial gene knockout mouse models with phenotypes in the cranial neural crest. Overall, this study is important for the field.However, I disagree with some secondary interpretations the authors give to their results. At the same time, the major conclusions stay solid. Below I discuss the most critical points.1) Chd7, Chd8 and Whsc1 are ubiquitously expressed. Thus, the specificity of regulation is achieved via interactions with other, more cell type- and stage-specific, factors. This would be good to mention.

We examined the in vivo expression of partners of TWIST1 and indeed they appear ubiquitously expressed in all neural crest populations extracted (Figure 4—figure supplement 2B, C). Their interaction would be subjected to the TWIST1 protein level across the subtypes of the neural crest.

2) The authors suggest: "The phenotypic data so far indicate that the combined activity of TWIST1-chromatin regulators might be required for the establishment of NCC identity. To examine whether TWIST1- chromatin regulators are required for NCC specification from the neuroepithelium and to pinpoint its primary molecular function in early neural differentiation, we performed an integrative analysis of ChIP-seq datasets of the candidates".This is a strange assumption, given that Twist1 is expressed only staring from the NCC delamination stage in mouse cranial neural crest (Soldatov et al., 2019). It does not seem to correlate with premigratory NCC identity and the situation inside of the neural tube. The authors conclude: "Therefore, combinatorial binding sites for TWIST1, CHD7 and CHD8 may confer specificity for regulation of patterning genes in the NECs." Or, alternatively, they may confer the control of mesenchymal phenotype, downstream migration and fate biasing etc. I do not think the authors have good arguments to bring up induction or patterning of NCCs at the level of neural tube.I have a good suggestion for the authors: I would extract the regulons from Soldatov et al. single cell data and run the binding site proximity check for the individual genes belonging to the gene modules /regulons specific to delamination and early NCC migration stages. I am curious, if the proximity of binding sites of Twist1-related crowd would rather correlate with genes from these specific regulons as compared to randomly selected regulons from the entire published single cell dataset.Randomization/bootstrapping analysis are welcomed. So far, being an excellent study, this paper does not solve a problem of downstream (of Twist1) gene expression program in the neural crest cells. At the same time, this is what the author can try to obtain with their DNA binding data in combination with published single cell data. Repression of Sox2 and upregulation of Pdgfra (reported in Figure 4) might be a part of this downstream program being in line with the published single cell gene expression data (Soldatov et al., 2019).The authors conclude the paragraph: "Therefore, combinatorial binding sites for TWIST1, CHD7 and CHD8 may confer specificity for regulation of patterning genes in the NECs". Again, this is not a good or plausible explanation based on specificity of expression of suggested pattering genes (or visualized genes are poorly selected). Additionally, although I believe the obtained results are important and of a good quality, I would not call them "developmentally equivalent to ectomesenchymal NCCs" or other NCCs. This is because the in vitro system will never reflect the embryonic in vivo development with high accuracy (especially when it comes to patterning and positional identity). This might explain that some prominent binding positions and interpretations the authors give do not correspond the gene expression logic during neural crest development. Besides, Twist1 and Chd7/8 are naturally expressed in many other cell types and might target non-NCC genes (Vegfa?). This does not reduce the value of the data, but it is good to mention for the community.

We thank the reviewer for bringing up these important points and for the helpful advice for further analysis. Our original data supported that in addition to the role of *Twist1*-CRM in the ectomesenchyme, where *Twist1* expression peaks, the interaction between *Twist1* and the chromatin regulators may have distinct functions at an earlier stage of NCC development. We have now studied the temporal profile of *Twist1* expression by mining the E8.5-10.5 NCC scRNA-seq data (Soldatov et al., 2019) and found that *Twist1* starts to be expressed in the delaminating cells. We have performed an integrative analysis of the single-cell RNA-seq data and our DNA binding data of TWIST1 and chromatin regulators to answer the following questions:

– At which stage/s does the module of *Twist1*-chromatin regulators kick into action, based on their expression across NCC subtypes and correlation with the DNA binding sites in the gene regulons specific for the NCC stage?

– What is the role of the TWIST1-module at different stages of NCC lineage development in vitro and in vivo?

The first clue came from examining the dynamics of expression of *Twist1* and partners. Similar to the outcome of in vitro differentiation model, the expression of *Twist1* and *Sox2* showed a reciprocal trend through the course of neuroepithelium to mesenchyme transition in vivo. *Twist1* initially peaks at the early migratory NCCs whereas, at more advanced stages, expression increases exponentially in the mesenchyme. On the other hand, as the reviewer pointed out, the three partners do not show major variations in expression levels during NCC differentiation and are expressed throughout differentiation (Figure 4—figure supplement 2A-C, subsection “Genomic regions co-bound by TWIST1 and chromatin regulators are enriched for early migratory NCC signatures in the open chromatin region”). Therefore, the expression level of *Twist1* is likely to be a major factor modulating the activities of TWIST1-CRM.

We then examined the activities of ChIP-seq targets of TWIST1, CHD7 and CHD8 in the scRNA-seq data, and specifically restricted the target sets to genes responsive to *Twist1* conditional knockout in *Wnt1-Cre; GFP* sorted E 9.0-9.5 mouse NCCs (Bildsoe et al., 2009). Among all the regulons in the developing NCC populations (E8.5- E10.5), the binding sites of TWIST1-module correlate best with the profile of early migratory NCCs. We also noted that the expression of marker genes of early migratory cells and neuroepithelium (neural tube) is mutually exclusive. A proportion of early migratory genes are significantly repressed in the neural tube (62%), while the neural tube genes are downregulated at the early migratory stage (33%). Therefore, TWIST1 and partners could repress many of the neural tube or neurogenesis genes in the early migratory NCCs, including *Sox2, Foxb1, Jag1, En1, Zic3* and *Dll3.*

In summary, combining the in vivo gene expression and DNA binding sites from NECs, *Twist1*- chromatin regulators are most active in the early migratory and mesenchyme stage. The mesenchyme only constitutes a small portion of cells in early neural crest development (E8.5). We will discuss the function of the module in the mesenchyme later combining the O9-1 knockdown data. Our observation suggests that even when the NCCs have delaminated from the neural tube, they still need to continuously repress many neurogenic genes. The TWIST1-module is likely a “valve” that maintains the one-way flow of newly specified NCCs towards downstream trajectories by preventing the flare-up of competitive neural stem cell regulators. Meanwhile, it helps to enhance/stabilize the migratory and early NCC fate programs.

The findings of the additional analysis were presented (see the aforementioned subsection) and discussed (Discussion) in the revised manuscript.

3) Figure 2: Twist1^-/+^ Chd8^-/+^ is repeated two times in panel B (but the embryos look differently), although the authors most likely meant to show Twist1^-/+^ Chd7^-/+^ in the second case. If this is indeed the case, the authors should also show a phenotype of Chd7 KO.

We have fixed the labeling issue. We did not have the resource to generate a *Chd7* KO. In this case, we compared the *Twist1^+/-^*; *Chd7^+/-^* with the *Twist1^+/-^* KO only to understand the genetic interaction.

4) The authors write: "Impaired motility in Twist1, Chd8 and Whsc1 knockdowns was accompanied by reduced expression of EMT genes (Pdgfrα, Pcolce, Tcf12, Ddr2, Lamb1 and Snai2) (Figure 6D, Figure 3—figure supplement 1D) and ectomesenchyme markers (Sox9, Spp1, Gli3, Klf4, Snai1), while 375 genes that are enriched in the sensory neurons located in the dorsal root ganglia (Ishii et al., 2012) were upregulated (Sox2, Sox10, Cdh1, Gap43; Figure 6E).– From the list of genes characterizing EMT, I can agree only on Pdgfra and Snai2, the rest is unspecific for EMT, and appears rather ubiquitous or specific to different cell populations (non-EMT).– From the list of suggested ectomesenchyme markers, I cannot pick any gene that would be a bit specific for ectomesenchyme (within neural crest lineage) except for Snai1. Sox9 is broadly expressed also in the trunk neural crest, Spp1 and Klf4 are not expressed in early mouse ectomesenchyme, Gli3 is too broad and non-selective. I suggest to select other gene sets (check the expression with online PAGODA app from Soldatov et al):http://pklab.med.harvard.edu/cgi-bin/R/rook/nc.p63-66.85-87.dbc.nc/index.html– The choice of DRG genes is also non-optimal, as Sox10 is pan-NCC, Sox2 is expressed in early migrating crest and satellite glial cells of DRG and Schwann cell precursors, Gap43 and Cdh1 are not specific enough. these gene clearly suggest the beginning of neuro-glial fates or trunk neural crest bias. To be more precise and for claiming sensory neurons, the authors should come up with pro-neuronal genes such as neurogenins, NeuroD, Isl1, Pou4f1, Ntrk and many others.Still, overall, I agree with author's main conclusions.

We agree with the reviewer that the selected markers are not the most typical ectomesenchyme or DRG markers. This was because the marker gene panel was designed to prioritize putative direct ChIP-seq target genes of the three factors. We have now performed qPCR analysis on additional markers highlighted in the scRNA-seq analysis of E8.5-E10.5 mouse NCCs (Soldatov et al., 2019). We focused on the analysis of genes associated with lineage bifurcation in the ectomesenchyme, autonomic and sensory branches, as these genes may best report trans-differentiating activities (http://pklab.med.harvard.edu/ruslan/neural.crest.html). Among these genes, we selected those harboring DNA binding sites for TWIST1-CRM in the regulatory elements (Figure 6—figure supplement 1).

Through qPCR analysis of the selected genes in O9-1 cranial NCC stem cells treated with individual or combinatorial siRNA, we studied the impact of the functional interaction of TWIST1 and its protein partners on the ectomesenchyme propensity versus the alternative lineages of NCCs. We detected the expression of most of the mesenchymal marker genes in the cell lines (Ct value < 32), but many sensory neurons and autonomic fate markers were below the detection level (Ct > 32). These initial observations suggested that the O9-1 neural crest stem cells may have a strong mesenchymal bias (Ishii et al., 2012). After 24 hours of siRNA knockdown of *Twist1, Chd8* and *Whsc1*, the treated cells displayed loss of mobility and reduced expression of EMT/ mesenchymal markers (Figure 6C, D). Most of the marker genes were significantly reduced in double siRNA treated groups. On the other hand, changes in non-mesenchymal genes, such as the sensory/ autonomic neuron markers, were inconsistent (Figure 6E). The significant upregulation of *Sox2, Sox8, Sox10* and *Tubb3* may indicate a gain of autonomic neuron fate, which is the immediately adjacent state of mesenchyme (Soldatov et al., 2019). Genes more specific for sensory neurons were either below detection or not significantly affected. Prolonged knockdown treatment may be required to observe a more substantial change of these alternative fate markers. This data agrees with observations in the *Wnt1-CreTwist1* conditional knockout mouse, where ectopic clumps of autonomic neurons expressing *Sox10, Phox2b* and *Ascl1* and *Tubb3* were found in the cardiac mesenchyme, while sensory markers (*TrkA, Pou4f1* and *NeuroD1*) were not altered (Vincentz et al., 2013); (discussed in more detail below). We therefore conclude that at the NCC maturation stage, high level of TWIST1 and its interactions with CHD8 and WHSC1 are important to commit cells to the mesenchymal lineage. Persistent activity of these factors is necessary to maintain the mesenchyme state and keep competitive neurogenic programs shut.

The findings were present in the Results, Figure 6C-E, and discussed (Discussion) in the revised manuscript.

5) The authors write: "The genomic and embryo phenotypic data collectively suggest a requirement of TWIST1- chromatin regulators in the establishment of NCC identity in heterogeneous neuroepithelial 403 populations". Again, I do not think the authors can claim anything related to the establishment of NCC identity. NCC identity, in broad sense, includes NCC induction within the neural tube, at both trunk and cranial levels. In mice, Twist1 is not expressed in trunk NCCs at all. At a cranial level, Twist1 is expressed too late to be a NCC inducing or patterning gene. As I mentioned earlier, it comes up during delamination.

We have modified the text to indicate that the initial function of TWIST1- chromatin regulators start from after delamination.

6) Figure 7G only partly corresponds to the positioning of the NCC markers in a mouse embryo. Id1 and Id2 are broadly expressed throughout all phases of NCC development and in entire dorsal neural tube beyond the NC region. Mentioning Otx2 as a NCC specifier is strange. At the same time, Msx1, Msx2, Zic1 are excellent genes! Tfap2 is a bit too late, but still ok. Please keep in mind, Msx1/2, Zic1 are expressed before Twist1, and, thus, Twist1 can be downstream of this gene expression program. Also, these genes become downregulated quite soon upon delamination, whereas Twist1/Chd7/8 expression stays (in vivo). Expression pattern of Tfap2a better corresponds to Twist1, although Tfap2a comes a bit before Twist1, and, besides, Tfap2a is expressed independently of Twist1 in trunk NCC. Despite such gene expression divergence, Twist1-based network might provide positive feedback loops stabilizing the expression of other transcriptional programs that were originally induced by other factors. It might be good mentioning this to the readers. This "stabilizing role" of Twist1 network can be a really important one. Given the incremental and combinatorial nature of the phenotype in vivo – this is most likely the case. I believe, these points are important to reflect in the Discussion section.

The “stabilizing role” summarizes the data more appropriately than the original interpretation. In the E8.5 single-cell tSNE map, the early NCC populations connect with the neural tube cells via the delaminating NCCs. *Twist1* comes in following cell delamination to enhance the expression of early NCC identity and repress the neural tube genes. This function of *Twist1* and partners appear to be specific to the rostral neural crest, as in *Twist1* mutants, the phenotype in the trunk neural crest is not pronounced (Chen and Behringer, 1995; Soo et al., 2002; Vincentz et al., 2008).

We added the reviewer’s suggestion in the Discussion, taking into account the expression timing of TWIST1-module and the NCC marker genes mentioned.

Reviewer #3:Using BioID, the authors identified more than 140 proteins that potentially interact with transcription factor Twist1 in a neural crest cell line. Most of these 140 Twist1-interactomes do not overlap with the 56 known Twist1 binding partners during neural crest cell development (see below). By focusing on several strong Twist1 binding partner candidates (particularly a novel candidate CHD8), the authors found:1) Twist1 interacts with these proteins via its N-terminal protein domain as demonstrated by co-IP.2) Compound heterozygous mutation of Chd8, Chd7 or Whsc and Twist1 displayed more severe phenotype compared to heterozygous mutation of Twist1 alone, for example, more significant reduction of the cranial nerve bundle thickness.3) ChIPseq analysis of Twist1 and CHD8 and key histone modifications revealed that the binding of Chd8 strongly correlate with those of Twist1, to active enhancers that are also labeled by H3K4me3 and H3K27ac.4) The binding of CHD8 requires the binding of Twist1, but not vice versa,5) Twist1-Chd8 regulatory module repress neuronal differentiation, and promotes neural crest cell migration, and potentially their differentiation into the non-neuronal cell types.The authors use an impressive array of different techniques, both in vitro and in vivo, and yield consistent results. The manuscript is nicely written. The findings are nuanced, but the major conclusions are largely expected.1) As the title states, the three key TWIST interacting factors that most of the study focuses on are chromatin regulators. However, the consequence of mutating these factors at the epigenetic level was not directly addressed, including the level of active histone modification, the accessibility of the Twist1/CDH co-bound promoters/enhancers, and the position of nucleosomes.

In the present study, we focussed on identifying the TWIST1 partners involved in neural crest differentiation and validating these interactions at the physical and functional level. The reviewer has made a good suggestion of follow up studies. Having found the most functionally relevant partners and the stage of lineage differentiation (early migratory and mesenchymal NCCs) that the TWIST1-CRM may act, we would be in the position to explore the epigenetic landscape through the analysis of histone modification, chromatin accessibility and target binding of these NCC lineage cell types in the mouse genetic models. This would be the scope of future studies.

2) CRISPR-generated ESCs and chimera technology were used effectively to generate mutants. In comparison, the analysis of the phenotypes was rather cursory and can benefit from more in-depth molecular analysis. Especially, the altered genes found in mutant NEC and NCC in the last section of the study should be validated in mutants.

We agree that the study would benefit from in-depth molecular analysis of the mouse models since all the chimeras were prioritized for phenotyping (by marker analysis) and thus not available for analysis of molecular pathway activity. While we are not able to generate more mutants at this juncture, we have undertaken an integrative analysis of our data in conjunction with scRNA-seq data of E8.5- E10.5 NCCs (Results) and highlighted the significance of changes in molecular activity in mutant NEC and NCC in vivo, by referencing reported phenotypes and transcriptome changes in *Twist1* knockout mutants (Discussion).

3) Across the manuscript, there were jumps from NCC to NEC and back. It will be important to justify why a certain cell type is selected for each analysis, focusing on the biological question at hand.

We have provided information on the use of cell types for specific analysis in the Results: ESC-derived NECs were used to study the early functions of *Twist1*-chromatin regulators, and neural crest stem cells were used as a surrogate of NCCs to study the function of module in fate bifurcation in post-migratory NCCs.

4) Using BioID, the authors detected 140 different proteins that interact with Twist1. However, only 4 of them overlap with the 56 known Twist1 partners (Figure 1A). This result suggests that BioID identified almost a distinct set of Twist1-interacting proteins, compared to the published results. The authors need to discuss the discrepancy, and the underlying reasons.

We listed the 56 know partners to make it clear about what has been reported for TWIST1 protein interaction and what are the novel partners identified in this study. It is important to note that the 56 known partners in the APID database are collected from various sources, including yeast two-hybrid and immunoprecipitation from mammalian cell lines. The lack of overlap is anticipated as protein interactions are highly cell-type-specific. Without high-throughput TWIST1 proteomic data, we cannot decide whether the BioID technique recovers a different group of protein compared with traditional immunoprecipitation techniques. However, the TCF3,4,12 partners that are recurrently found in yeast-two-hybrid, immunoprecipitation and in vitro interaction assays were recovered by BioID (El Ghouzzi et al., 2000; Firulli et al., 2007; Fu et al., 2011; Teachenor et al., 2012; Sharma et al., 2013; Kotlyar et al., 2015; Li et al., 2015). This has encouraged us to explore the rest of the novel partners found in the BioID data.

These points are noted in the Results.

5) The authors show that Twist1 colocalize with Cdh8, and is required for the binding of Cdh8, thus suggest that Twist1-Cdh8 form a regulatory module. Given the degenerate nature of bHLH factor binding motifs, it is likely that the binding of Twist1, and subsequently the binding of Cdh8, are dictated by other transcription factors. Therefore, a motif enrichment analysis should be done among the Twist1/Cdh8 co-binding sites, and compare those motifs enriched in Twist1-only and Cdh8-only binding sites.

We agree with the reviewer that bHLH factor binding sites alone is not enough to confer specificity for TWIST1-CHD8 target recognition. The specificity for NCC differentiation genes might be achieved when TWIST1, CHD8 and additional factors bind adjacently to each other, either sequentially or simultaneously.

We performed the suggested analysis of motif enrichment of three ChIP-seq peak groups (TWIST1+CHD8, CHD8-only and TWIST1 only). The top motifs and best-matching factors binding to these motifs are now shown in Figure 4—figure supplement 3A. These factors could be either co-binding partners or competitors for the same region. Interestingly the three groups of peaks were enriched in a distinct range of motifs. The co-binding/ competitive factors identified in the three groups of peaks also had variable express patterns across the NCC subtypes. While the average expression of factors binding to TWIST1+CHD8 peaks was activated in the delaminatory and all migratory NCC populations, those binding to CHD8-only and TWIST1-only peaks were highly expressed in the neural tube (Figure 4—figure supplement 3B). This analysis suggests that the TWIST1-CHD8 module is interacting with additional factors, including DLX1 and SOX10 in this case, to enhance NCC identity (Discussion).

6) The increasing expression of DRG neurons genes in Twist1/Cdh8 mutants suggests a possible transition from cranial NC to trunk NC. Therefore, the authors should examine the expression of marker genes accordingly.

We have selected a representative panel of ectomesenchymal genes and markers of autonomous and sensory NCC fate in the trunk NCCs. As discussed in the response to reviewer #1 comment 4 above, we observed a loss of ectomesenchymal propensity and some gain of autonomic neuron potential revealed by the upregulation of *Sox2, Sox8, Sox10* and *Tubb3* in the compound mutant embryos. However, genes specific for sensory neurons either were below detection or not significantly changed. These changes were consistent with the ectopic expression of autonomic neuron marker in *Twist1* conditional knockout mouse (Vincentz et al., 2013). We concluded that the *Twist1*/*Chd8*/*Whsc1* module influences the mesenchymal bias at the autonomic**-**ectomesenchyme bifurcation. The findings were present in the Results, Figure 6C-E, and discussed (Discussion) in the revised manuscript.